

# The off-line Lagrangian particle model FLEXPART-NorESM/CAM (V1): model description and comparisons with the on-line NorESM transport scheme and with the reference FLEXPART model.

Massimo Cassiani[1], Andreas Stohl[1], Dirk Olivié[2], Øyvind Seland[2], Ingo Bethke[3], Ignacio Pisso[1], Trond Iversen[2].

[1]NILU – Norwegian Institute for Air Research, P.O. Box 100, 2027 Kjeller, Norway
[2]Norwegian Meteorological Institute, P.O. Box 43, Blindern, 0313 Oslo, Norway.
[3]Uni Research Climate, Bjerknes Centre for Climate Research, Bergen, Norway, P.O. Box 7810, 5020 Bergen, Norway.

*Correnspondence to:*Massimo Cassiani (mc@nilu.no)

**Abstract.** The off-line FLEXible PARTicle stochastic dispersion model (FLEXPART) is nowadays a community model used by many scientists. Here, an alternative FLEXPART model version has been developed, tailored to use with the meteorological output data generated by the CMIP5-version of the Norwegian Earth System Model (NorESM1-M). The atmospheric component of NorESM1-M is based on the Community Atmosphere Model (CAM4), hence this FLEXPART version could be widely applicable and it provides a new advanced tool to directly analyse and diagnose atmospheric transport properties of the state-of-the-art climate model NorESM in a reliable way. The adaptation of FLEXPART to NorESM required new routines to read meteorological fields, new post-processing routines to obtain the vertical velocity in the FLEXPART coordinate system and other changes. These are described in detail here. To validate the model, several tests were performed that offered the possibility to investigate some aspects of off-line global dispersion modelling. First, a comprehensive comparison was made between the tracer-transport from several point sources around the globe calculated on-line by the transport scheme embedded in CAM4 and the FLEXPART model applied off-line on output data. The comparison allowed investigating several aspects of the transport schemes including: the approximation introduced by using an off-line dispersion model with the need to transform the vertical coordinate system, the influence on the model results of the sub-grid scale parameterizations of convection and boundary layer height and the possible advantage entailed in using a numerically non-diffusive Lagrangian particle solver. Subsequently, a comparison between the reference FLEXPART model and the FLEXPART-NorESM/CAM version was performed to compare the well-mixed state of the atmosphere in a one-year global simulation. The two model versions use different methods to obtain the vertical velocity but no significant difference in the results was found. However, for both model versions there was some degradation in the well-mixed state after one-year of simulation. Finally, the capability of the new combined modelling system in producing realistic backward in time transport statistics was evaluated calculating the average footprint over a five-year period for several measurement locations and by comparing the results with those obtained with the reference FLEXPART model driven by re-analysis fields. This comparison confirmed the effectiveness of the combined modelling system FLEXPART with NorESM in producing realistic transport statistics.



# 1    Introduction

Transport in the atmosphere can be simulated with grid based methods or Lagrangian particle methods. A distinct advantage of Lagrangian particle models is that they are essentially free of numerical diffusion errors (except for errors associated with interpolations), whereas Eulerian and semi-Lagrangian grid based methods normally suffer from numerical diffusion (see e.g.

Reithmeir and Sausen, 2002). This limits, for instance, the capabilities of Eulerian models to simulate intercontinental pollution transport (Rastigejev et al., 2010). Purely Lagrangian transport schemes become an especially attractive option when the focus is on tracer transport rather than on atmospheric chemistry. In this case, the Lagrangian scheme naturally simulates only the domain of interest achieving a high computational efficiency with no compromises in the spatial accuracy. Lagrangian particle models used nowadays for atmospheric transport are mostly based on stochastic approaches to describe unresolved fluctuating

motions of the particles in the atmosphere. This class of models is referred to as Lagrangian stochastic (LS) models (see e.g. Thomson, 1987; Stohl et al., 1998; Draxler, 1999; Luhar and Hurley, 2003; Lin et al., 2003; Jones et al., 2004; Rossi and Maurizi, 2014). A nice feature of off-line Gaussian LS models is that they can be run backward in time without any model changes other than the sign changes of wind components (e.g. Thomson, 1987; Flesch et al., 1995; Stohl et al., 2003; Seibert and Frank 2004). Furthermore, atmospheric turbulence (in the boundary layer) can be treated more accurately in LS particle

models than in grid-based dispersion models. All three features – minimal numerical diffusivity, possibility of time-reversed transport, and accurate turbulence description – are particularly attractive for atmospheric inversion studies, where sources of emissions (e.g., of greenhouse gases) are determined by combining information from atmospheric measurements and dispersion models. Not surprisingly, LS models are popular tools for such studies (e.g., Gerbig et al., 2003; Thompson and Stohl, 2014, Henne et al. 2016).

Therefore, and for many other purposes, many different off-line Lagrangian models have been developed, probably the most popular being the Hybrid Single-Particle Lagrangian Integrated Trajectory (HYSPLIT) model (Draxler, 1999), the Stochastic Time-Inverted Lagrangian Transport (STILT) model (Lin et al., 2003), and the FLEXible PARTicle stochastic dispersion model (FLEXPART) model (Stohl et al., 1998). In this paper, we concentrate on the FLEXPART model. The reference version of FLEXPART (Stohl et al., 1998, 2005; Stohl and Thomson, 1999) uses global meteorological data from the European Centre

for Medium-Range Weather Forecasts (ECMWF) or the National Center of Environmental Prediction (NCEP). It is a versatile tool that has been applied in many different fields of atmospheric research ranging from classical pollution dispersion modelling to measurement data interpretation studies, inverse modelling, or studies of the hydrological cycle.

FLEXPART has become a community model. Scientists from several countries contribute to its development and share model versions and model branches on a website (https://www.flexpart.eu/) based on Trac (http://trac.edgewall.org/), which is an

enhanced wiki and issue tracking system for software development projects. The git (https://git-scm.com/) version control system is used. Most notably, a version adapted for the WRF (Weather Research and Forecasting) model has been documented extensively (Brioude et al., 2013). This version takes advantage of several different coordinate systems supported by WRF.



The current paper describes a new branch of FLEXPART that uses output data from the Norwegian Earth System Model (NorESM1-M; Bentsen et al., 2013; Iversen et al., 2013), which is based on the Community Climate System Model (CCSM4, Gent et al., 2011; Vertenstein et al., 2010). In NorESM, the Community Atmosphere Model (CAM4, Neale et al. 2010) is modified to include the aerosol module developed for NorESM (CAM4-Oslo, Seland et al., 2008; Kirkevåg et al., 2013). This

version of FLEXPART is named FLEXPART-NorESM/CAM, and is tailored particularly to climate applications and adjusted to use the NorESM1-M and CAM4 output coordinate system and data formats.

In section 2, we introduce FLEXPART-NorESM/CAM while a description of relevant aspects of NorESM1-M and CAM4-Oslo is reported in Appendix A and some technical details of the model coupling are reported in Appendix B. In section 3, we validate FLEXPART-NorESM/CAM as a tool for transport diagnostics, by comparing its Lagrangian off-line tracer dispersion

calculations with the finite volume on-line tracer calculations used in CAM4. The comparison includes tracer releases from several point sources around the globe, and it allows for evaluating: i) the correctness of this FLEXPART version, ii) the differences introduced by the need to transform the vertical coordinate system and obtain an appropriate vertical velocity, iii) the use of two methods to obtain the vertical velocity in FLEXPART-NorESM/CAM, iv) the influence on the models results of different sub-grid scale (SGS) parameterizations for convection and boundary layer height and v) the possible advantages

entailed in using a numerically non-diffusive Lagrangian particle solver instead of a grid-based solver. The maintenance of the well-mixed state (e.g. Thomson, 1987) of the particles is also investigated in section 3, comparing the results of the reference FLEXPART model and the new version in a one-year global simulation of the whole atmosphere up to about 20km above ground. Such a global scale well-mixed test was not previously done, to our knowledge, for an off-line Lagrangian stochastic dispersion model, and it is relevant in case there is need to simulate long-term evolution of the whole atmosphere and eventually

include chemistry and mixing in a Lagrangian framework (e.g. Collins et al. 1997, Reithmeir and Sausen, 2002, Stenke et al. 2009). In section 4, we compare climatological FLEXPART-NorESM/CAM calculations with equivalent calculations done with the reference FLEXPART model version driven with atmospheric re-analysis data to test the effectiveness of the combined modelling system FLEXPART-NorESM in producing realistic transport statistics. Finally, in section 5, we draw conclusions.

## 25  2   Model description

FLEXPART-NorESM/CAM has been developed on the basis of FLEXPART version 9.1, which can be used with meteorological data from ECMWF or NCEP. No detailed separate documentation of FLEXPART version 9.1 exists but the code is available from https://flexpart.eu/, and earlier versions were described by Stohl et al. (1998, 2005). The most recent FLEXPART user guide is available at https://www.flexpart.eu/downloads/26, and an up-to-date model description is under

development. In section 2.1, we will give a short overview of some salient features of the FLEXPART model that are important for the development of the FLEXPART-NorESM/CAM version. In section 2.2, the changes to FLEXPART introduced with this NorESM/CAM-version of the model will be explained; most notably to allow for a different vertical coordinate system in



the input meteorological fields. In Appendix A, the NorESM1-M is briefly introduced, emphasizing the atmospheric component CAM4-Oslo and giving the details of the model settings used to run the current simulations. For brevity in the remainder of the manuscript we will refer to CAM4-Oslo and NorESM1-M simply as CAM and NorESM. In Appendix B the FLEXPART-NorESM/CAM input data structure and name of variables are given in detail.

## 2.1 Brief description of the original FLEXPART model

The FLEXPART model is coded in Fortran 95. The physics of the FLEXPART model is described in detail in Stohl et al. (2005), although since then many improvements have been introduced. FLEXPART is a Lagrangian stochastic particle model and uses Thomson's (1987) well-mixed criteria and diffusion coefficients to define stochastic differential equations for the motions of notional fluid particles. The stochastic components simulate the effects of the planetary boundary layer (PBL) turbulence and unresolved mesoscale motions. We underline that the model for the PBL accounts for different stabilities, including possibility of skewed turbulence in convective conditions, and for the vertical air density gradient (Stohl and Thomson, 1999, Cassiani et al. 2015). More details of the PBL turbulence parametrizations can be found in the FLEXPART user's manual. FLEXPART simulates deep moist convective exchanges using a non-local transilient transport matrix constructed consistently with the scheme of Emanuel and Živković-Rothman (1999) (see Forster et al., 2007), and includes dry/wet deposition processes and linear chemical and radioactive loss processes. Overall, the Lagrangian representation used in the model does not have significant numerical diffusivity, and preserves well the structures generated during the advection and dispersion processes and this will also be shown in detail below and compared to the finite volume solver of NorESM. A grid or weighting kernel is used only to extract statistically averaged information from the particles at the required spatial resolution. The model can run both in forward or backward mode to study dispersion from a source or receptor respectively (e.g. Thomson 1987, Flesch et al. 1995, Stohl et al., 2003).

FLEXPART uses a terrain following vertical coordinate system $\tilde{z} = z - z_g$, where $z$ is the Cartesian vertical coordinate and $z_g$ the model topography . The vertical velocity in the terrain following coordinate is obtained by post-processing the velocity field provided by the ECMWF model. In the ECMWF model the vertical velocity, $\dot{\eta}$ ($s^{-1}$), is expressed in a generalized discrete hybrid coordinate system defined as (see e.g. Simmons and Burridge 1981, Untch and Hortal, 2004, or the FLEXPART user guide)

$$\eta_k = \frac{A_k}{P_0} + B_k \tag{1}$$

where $k$ indicates a vertical model level, $A_k$(Pa) and $B_k$(unitless) are coefficients and $p_0$ is a constant reference pressure of 1013.25 hPa. The transformation used in the FLEXPART model to obtain the terrain following velocity from the $\dot{\eta}$ velocity in the hybrid system is:



$$\widetilde{w} = \dot{\eta}\left(\frac{\partial p}{\partial \eta}\right)\left(\frac{\partial \tilde{z}}{\partial \tilde{z}}\right)^{-1} + u\left.\frac{\partial \tilde{z}}{\partial x}\right|_{\eta} + v\left.\frac{\partial \tilde{z}}{\partial y}\right|_{\eta} \tag{2}$$

where $p$ is the static pressure, $\widetilde{w} = d\tilde{z}/dt\,(\text{m s}^{-1})$ the vertical velocity in the terrain following coordinate system, and $u$ and $v$ the mean wind horizontal velocity components in the east-west and north-south directions respectively.

## 2.2 FLEXPART-NorESM/CAM vertical velocity, ten meters wind and dew point

From the original NorESM and CAM model output fields, a few more fields necessary to run the model, are obtained by on-line post processing: the vertical velocity, the 10-m wind and the dew point.

The vertical velocity provided by NorESM and CAM output is $\varpi = dp/dt\,(\text{Pa s}^{-1})$. In FLEXPART-NorESM/CAM two methods are available to obtain the vertical velocity in terrain following coordinates, of which one can be chosen by setting an extra flag in the file COMMAND (see Appendix B). The first method does not use explicitly the hydrostatic assumption while the second method does.

From the definition of total derivative in the hybrid $\eta$ coordinate system (see e.g. Simmons and Burridge 1981) we have,

$$\omega = \frac{\partial p}{\partial t} + u\frac{\partial p}{\partial x} + v\frac{\partial p}{\partial y} + \dot{\eta}\frac{\partial p}{\partial \eta} \tag{3}$$

From this definition, the quantity $\dot{\eta}\,\partial p/\partial \eta$, which is needed in Eq. (2) to define $\widetilde{w}$, can be obtained on the eta hybrid levels of NorESM1-M as,

$$\dot{\eta}\left(\frac{\partial p}{\partial \eta}\right) = \varpi - \left.\frac{\partial p}{\partial t}\right|_{\eta} - u\left.\frac{\partial p}{\partial x}\right|_{\eta} - v\left.\frac{\partial p}{\partial y}\right|_{\eta} \tag{4}$$

Equation (4) has been discretized with a simple finite difference scheme in space and time,

$$\dot{\eta}\left(\frac{\partial p}{\partial \eta}\right)_{i,j,k,t} = \varpi_{i,j,k,t} - \frac{1}{2\Delta t}\left(p_{i,j,k,t+\Delta t} - p_{i,j,k,t-\Delta t}\right) - \frac{1}{2\Delta x}u_{i,j,k,t}\left(p_{i+1,j,k,t} - p_{i-1,j,k,t}\right)$$
$$- \frac{1}{2\Delta y}v_{i,j,k,t}\left(p_{i,j+1,k,t} - p_{i,j-1,k,t}\right) \tag{5}$$

with $i, j$ indicating a grid point in the direction of longitude and latitude respectively, $k$ indicating a vertical level (constant $\eta$) and $t$ the current time and $\Delta x$, $\Delta y$ and $\Delta t$ are space and time discretization intervals. Note that an initial field at time $t - \Delta t$ is





necessary for the discretization in time. Therefore, with this method to obtain the vertical velocity, a FLEXPART simulation can only start at the time of the second available NorESM/CAM output field. At the poles, the vertical velocity value is obtained from the space average of the neighbourhood values (one grid point away from the poles), in the same way as done in the original FLEXPART-ECMWF model. Afterwards, the values of $\dot{\eta}\left(\frac{\partial p}{\partial \eta}\right)$ are linearly interpolated to the terrain-following

FLEXPART levels where they are used (together with the other interpolated terms in Eq. (2)) to obtain the terrain following velocity $\widetilde{w}$. Note that at the ground the value $\widetilde{w} = 0$ is assigned. For points in the FLEXPART terrain-following coordinate system that are located below the first NorESM layer above ground, vertical velocity is linearly interpolated between vertical velocity at this first layer and the $\widetilde{w}(= 0)$ value assigned at the ground. The number of vertical terrain-following levels used in FLEXPART-NorESM/CAM is the same as the number of hybrid levels used in NorESM/CAM, and therefore the positions

of the lowest grid points above ground are very close to each other in the two coordinate systems.

The second method uses explicitly the hydrostatic approximation, i.e. $\partial p/\partial z = -\rho g$ , where $\rho$ is the air density and $g$ the gravitational acceleration, to obtain the vertical velocity and therefore,

$$\widetilde{w} = -\frac{\omega}{\rho g} - u\frac{\partial z_g}{\partial x} - v\frac{\partial z_g}{\partial y} \tag{6}$$

The vertical velocity $w = -\frac{\omega}{\rho g}$ is first obtained at the original $\eta$ levels and subsequently interpolated to the FLEXPART-

NorESM/CAM terrain following levels, where the correction for the topography is applied. The assignment of the velocity at the ground and the poles is as discussed above. The two methods give almost indistinguishable results and this will be shown below.

Other minor modification necessary to use FLEXPART with the wind fields generated by NorESM include a procedure to obtain the ten meters wind components and the dew point. The 10-m wind components are obtained from the 10-m wind

velocity module and the surface stresses as follows:

$$u|_{\tilde{z}=10m} = (u^2 + v^2)^{\frac{1}{2}}\Big|_{\tilde{z}=10m} \times \frac{\overline{u'w'}}{\left(\overline{u'w'}^2 + \overline{v'w'}^2\right)^{\frac{1}{2}}}\Bigg|_{\tilde{z}=0m}$$

$$v|_{\tilde{z}=10m} = (u^2 + v^2)^{\frac{1}{2}}\Big|_{\tilde{z}=10m} \times \frac{\overline{v'w'}}{\left(\overline{u'w'}^2 + \overline{v'w'}^2\right)^{\frac{1}{2}}}\Bigg|_{\tilde{z}=0m} \tag{7}$$



In Eq. (7) the prime denotes a fluctuation from the resolved mean quantity and the overbar is the averaging operator, therefore $\overline{u'w'}$ and $\overline{v'w'}$ are the north-south and east-west surface stresses.

The dew point is obtained as (see e.g. Campbell and Norman 1998, p. 43)

$$T_d = \frac{C \ \ln\left(\frac{e}{A}\right)}{B - \ln(\frac{e}{A})} \tag{8}$$

where $A = 611 \text{Pa}$, $C = 240.97°C$ and $B = 17.502$, for temperatures above $0°C$ and $C = 265.5°C$ and $B = 21.87$ for temperatures below $0°C$, $e$ is the ambient partial pressure of water vapour obtained from specific humidity and surface pressure.

## 3    Model validation and comparison

Despite this model version being similar to the reference FLEXPART model, several changes have been introduced. These changes needed testing to verify the correctness and performance of the new model and created also the opportunity for models comparisons. First, the results obtained by the on-line advection and diffusion equation solver in NorESM and the off-line Lagrangian particle tracking and diffusion used in FLEXPART-NorESM/CAM were compared in numerical tracer experiments. In doing this, we investigated the effects on the results of the change of vertical coordinate system, SGS convection scheme and PBL depth parametrization, which are different between the off-line FLEXPART-NorESM/CAM and the on-line transport scheme of NorESM. Furthermore, with the aim of validating the procedure used to obtain the vertical velocity, and the possible numerical errors introduced by this procedure, a comparison between the vertical well-mixed state in FLEXPART-NorESM/CAM and the reference FLEXPART-ECMWF has been performed for a one-year global domain-filling  simulation of the atmosphere. To our knowledge such quantitative and comparative tests on the vertical well-mixed state were never before presented in the literature for a global-scale off-line Lagrangian particle model over a one year simulation.

### 3.1    Off-line FLEXPART-NorESM/CAM versus on-line NorESM scalar transport

In this section we compare the results for scalar tracer transport obtained using the NorESM on-line grid-based advection and diffusion equation solver (see Appendix A) and the results obtained off-line using the fully Lagrangian FLEXPART-NorESM/CAM.  An integrated solver is generally more consistent with the dynamics of the model than a post-processor, which moreover in the present case works with a different coordinate system (see e.g. Byun, 1999). Furthermore, on-line tracer advection and diffusion in NorESM is calculated with the model's 30-minute time step, whereas FLEXPART-NorESM/CAM is driven off-line with 3-hourly NorESM output fields. FLEXPART uses here a 5-minutes time step for the integration of the stochastic differential equations for particle velocities with a refinement to 75 seconds for the fluctuating vertical velocity



component in the PBL. However, resolved scale processes, numerical diffusion and parametrized processes all influence the modelled dispersion of a tracer in the atmosphere. In this respect Lagrangian particle tracking has some advantages compared to a grid-based solver including negligible numerical diffusivity and a much better framework for including turbulence parametrizations especially in the PBL (e.g. Thomson, 1987, Stohl and Thomson, 1999, Cassiani et al., 2015). SGS

parametrizations of PBL turbulence and depth, topographical effects on PBL depth, mesoscale horizontal motions and moist convection are all important for tracer dispersion, and they are different in the on-line NorESM transport scheme and the off-line FLEXPART-NorESM/CAM. Thus, while it may not be clear which transport scheme performs better overall, it is important to ensure the overall consistency of the two schemes.

For comparisons of the two transport schemes, tracer dispersion from fifteen sources on the globe was calculated using both

NorESM on-line transport scheme, and FLEXPART-NorESM. Passive tracers were released from 1.89° × 2.5° (latitude × longitude) boxes (i.e., within NorESM grid boxes) and at the ground both continuously and almost instantaneously (over one NorESM model time step of 30 minutes). At the poles a disk of 0.95° radius was used. The release started at 00:00:00 (UTC) of January 2nd, 2011. The results are shown after 144 hours from the start of the release. The list of sources and their location is reported in Table 1 but only few salient results are shown here. In NorESM all the parametrizations were active while in

FLEXPART-NorESM/CAM, some parametrizations were switched off in turn, for sensitivity studies. In the following comparisons, the results of NorESM are presented (linearly interpolated) in the terrain following coordinate system of FLEXPART.

| Short name | Location | | | Full name |
|---|---|---|---|---|
| | Lon. | Lat. | Elevation(m) | |
| NPL | 0.0 | 90.00 | 0 | North pole |
| SPL | 0.0 | -90.00 | 2850 | South pole |
| ALE | 297.5 | 82.42 | 200 | Alert station |
| BAR | 202.5 | 71.05 | 77 | Barrow station |
| SUM | 322.5 | 72.95 | 2890 | Summit station |
| ZEP | 12.5 | 78.63 | 116 | Zeppelin station |
| TRO | 2.5 | -72.95 | 2185 | Troll station |
| HNA | 355.0 | 78.63 | 0 | High North Atlantic |
| MNA | 325.0 | 48.32 | 0 | Mid North Atlantic |
| EQA | 335.0 | 0.95 | 0 | Equator Atlantic |
| MSA | 345.0 | -40.74 | 0 | Mid South Atlantic |
| ALP | 10.0 | 48.32 | 845 | Alps |
| BOR | 115.0 | 2.84 | 416 | Borneo |
| GUI | 145.0 | -6.63 | 734 | New Guinea |
| HIM | 90.0 | 29.37 | 4656 | Himalaya |

**Table 1: List of point source locations used in the performed model test.**

We begin our analysis with the results obtained for a continuous release at the ALP location. Here, by switching on and off

the parametrization, SGS parametrization of topography was tested to have a minor role and the moist convection scheme was



found as well to have some but not a dominant influence. Therefore, resolved scale motions dominate this plume dispersion. The topography at the source is also intermediate among the examined cases. Figure 1 shows the vertically integrated and meridionally integrated tracer concentrations after 144 hours of simulation. For FLEXPART-NorESM/CAM results obtained with both the methods to obtain the vertical velocity are shown. First, we note that the two methods to obtain the vertical

velocity produce almost indistinguishable results. The tracer fields by NorESM and FLEXPART-NorESM/CAM are also generally very similar, and the major differences can be attributed to the higher numerical diffusivity of the on-line finite volume based solver, a point that will be further investigated in section 3.1.4. The purely Lagrangian particle solver shows the characteristic ability to maintain a sharper gradient (e.g. Reithmeir and Sausen, 2002) and preserve filamentary structures that are generated by chaotic advection (Ottino, 1989) which grid-based models have difficulties to capture. The results shown in

Fig. 1 can be considered typical, i.e. with no extreme topography and without substantial influence from SGS parametrizations. Below, we will investigate cases that allow highlighting particular aspects of tracer transport.

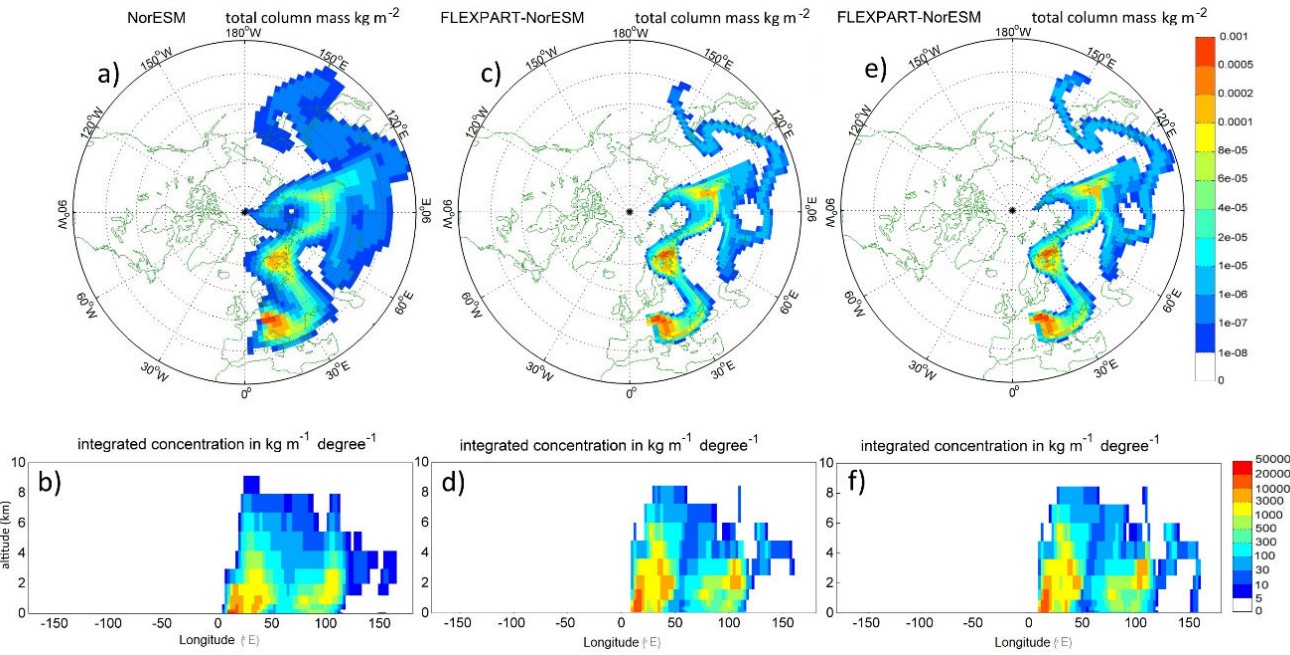

**Figure 1. Tracer dispersion for a continuous release from the ALP source, with the NorESM on-line tracer solver (a,**
**b) versus Lagrangian tracking with FLEXPART-NorESM/CAM, with vertical velocity obtained with the method one (c, d) and the method two (e, f). The panels a, c and e show maps of vertically integrated tracer concentrations, the panels b, d and f show meridionally integrated tracer concentrations as a function of longitude and altitude, 144 hours after the start of the simulation. Notice that the colour scales used span several orders of magnitude and emphasize differences between the two simulations that would not be visible on a linear scale.**





### 3.1.1 Consistency of results after transformation of vertical coordinates

Here we examine the results obtained for a continuous release in the SUM location. This release point is over high topography where the transformation of the vertical velocity has a significant role and can potentially create larger errors than over flat and low terrain. Moreover, this location was selected since the roles of SGS parametrization of topography and convection

were found (by switching on and off the parametrizations in separate tests; not shown) to be completely negligible in the FLEXPART-NorESM/CAM simulations. Although the same test was not performed for the NorESM native solver, it is reasonable to assume that the role of the convection scheme was minor also in that case. Therefore, for this case the SGS physical parametrizations can be considered negligible while at the same time the transformation of vertical coordinate is very important. In Fig. 2, the vertically integrated and meridionally integrated tracer concentrations are shown after 144 hours of

simulation. It is possible to see again that the two methods to obtain the vertical velocity in FLEXPART-NorESM/CAM produce almost indistinguishable results. The tracer fields simulated by NorESM and FLEXPART-NorESM/CAM are very similar, the difference being even smaller than in the previous (ALP) case examined, and this shows that in general the different model parametrizations introduce some deviations in the results, while the differences induced by the transformation of vertical velocity are in comparison of minor importance. The effects of parametrizations will be further investigated below in sections

3.1.2 and 3.1.3. In this SUM case the effect of the higher numerical diffusivity of the grid-based on-line solver is even clearer than in the ALP case.

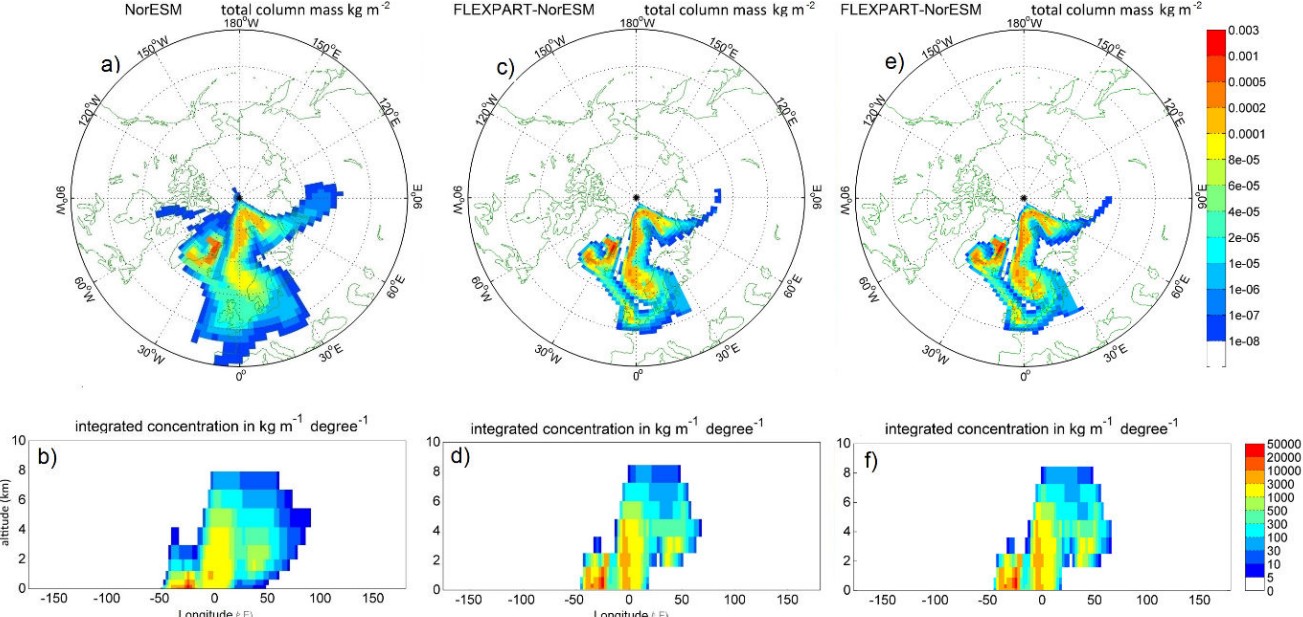

**Figure 2. As in Fig. 1 but for SUM.**





A further test of the correctness of the transformation of vertical velocity over topographic slope regards the HIM release point, which is over the highest NorESM grid point of the whole domain. By switching on and off the convection scheme in FLEXPART-NorESM/CAM, we have found that the convection scheme in this case has a minor but not completely negligible influence, similarly to the ALP case investigated above. In any case, the agreement between NorESM and FLEXPART-
NorESM/CAM is again very good, with differences mainly generated by numerical diffusivity. The results of FLEXPART-NorESM/CAM when using the two different methods to obtain the vertical velocity were again almost indistinguishable hence in Fig. 3, and in the remainder of the paper, only results obtained using the first method are shown.

Overall, for both these cases of high topography the agreement between the on-line and off-line transport schemes is remarkable. These initial tests grant confidence that both the procedures used to obtain the vertical velocity in the FLEXPART
terrain following coordinate are appropriate and do not add any significant errors and that the use of an off-line transport scheme is adequate. Moreover, the Lagrangian solver is able to maintain better the filamentary structure generated by chaotic advection.

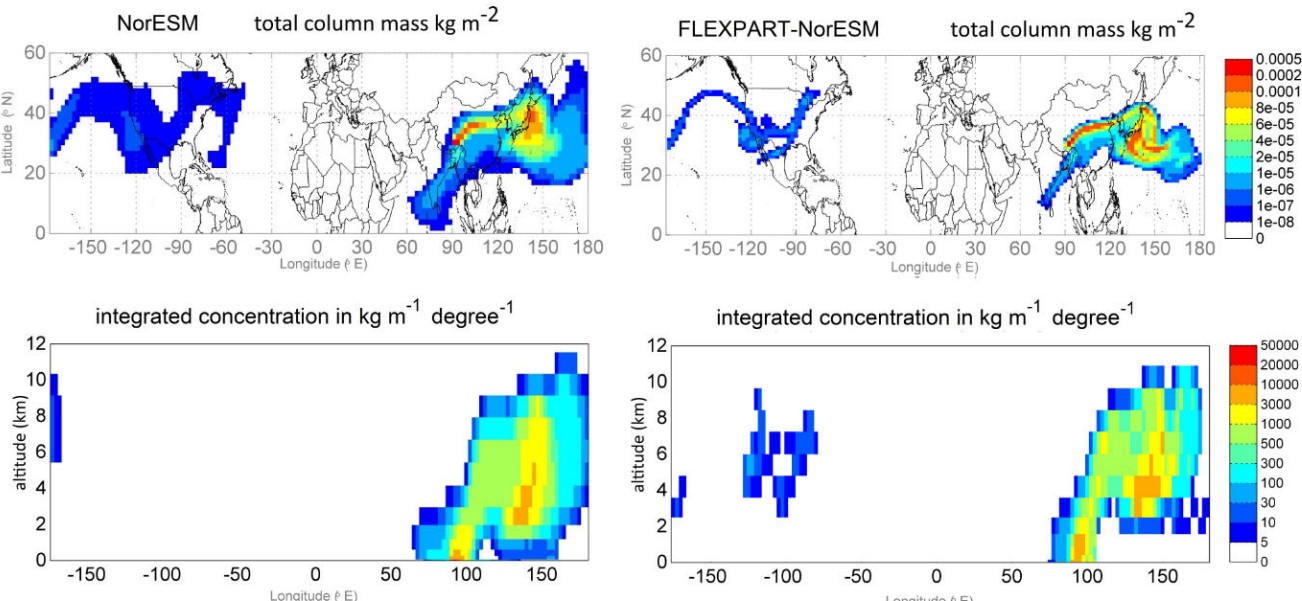

**Figure 3. Tracer dispersion for a continuous release from the HIM source, with the NorESM on-line tracer solver (left**
**panels) versus Lagrangian tracking with FLEXPART-NorESM/CAM, with vertical velocity obtained with the method one (right panels). Upper panels show maps of vertically integrated tracer concentrations, the lower panels show meridionally integrated tracer concentrations as a function of longitude and altitude, 144 hours after the start of the simulation.**

### 3.1.2    Consistency between moist convection parametrization in NorESM and FLEXPART-NorESM/CAM

In model runs with horizontal resolution of 1.89° × 2.5° used for the current NorESM simulations, the convection parametrization plays a major role for momentum and mass vertical exchange.  As outlined in the technical note describing



CAM4 (http://www.cesm.ucar.edu/models/ccsm4.0/cam/docs/description/cam4_desc.pdf), NorESM uses a modified Zhang and McFarlane (1995) convection scheme for momentum and tracer transport, while FLEXPART uses the Emanuel and Živković-Rothman (1999) parametrization. It is therefore instructive, and a further general validation for FLEXPART, to briefly investigate the consistency between these two parametrizations and their implementation in the models. For doing this,

we selected a location where the convection parametrization plays a dominant role in vertical dispersion, EQA, which is located at the equator at the sea level and over sea. For this location, neither the sub-grid scale topography parametrization nor the vertical velocity transformation plays any significant role.

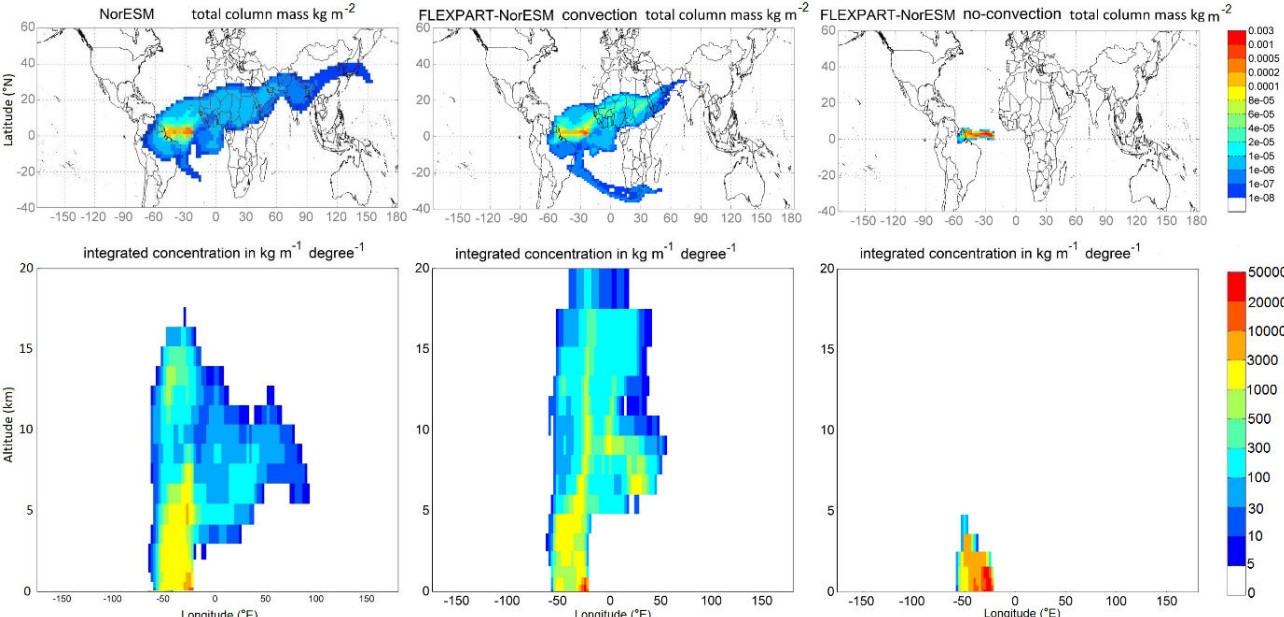

**Figure 4. Continuous release from the EQA source. FLEXPART-NorESM/CAM results without convective scheme**
**(right panels) and with convective scheme (central panels) versus NorESM integrated solver results (left panels) are shown. The upper panels show maps of vertically integrated tracer concentrations, the lower panels show meridionally integrated tracer concentrations as a function of longitude and altitude, 144 hours after the start of the simulation.**

As can be seen in Fig. 4, comparing right and central panels, the deep convection scheme in this case totally alters the vertical (and consequently also the horizontal) distribution of the tracer by lifting it up to 20 km compared with an uplifting of less

than 5 km without it. The results of FLEXPART-NorESM/CAM with the convection activated and NorESM are quite similar, despite the different convection schemes used. However, the convection scheme in FLEXPART transports a larger mass fraction above about 7.5 km, which may be due to simulation of overshooting convection with the Emanuel and Živković-Rothman (1999) parametrization. This in turn generates a faster horizontal advection with higher total column integrated concentration extending over the central region of the African continent.

For all the tropical sources, the convection scheme dominates the dispersion and the relative behaviour of the two models in all these cases is similar, but not perfectly consistent. For example, Fig. 5 shows the comparison for a short (30 minutes) release



in the GUI location after 144 hours of dispersion. As in the EQA case a higher overall plume vertical extension in FLEXPART-NorESM/CAM with respect to NorESM can be observed. However, in the GUI release the mass fraction transported above about 7.5 km in FLEXPART-NorESM/CAM is lower than that of NorESM. The overall agreement is good, considering that different convection schemes are used, but some discrepancies can be observed. Near the east coast of Australia and north-

west of Madagascar FLEXPART-NorESM/CAM simulates significantly higher column integrated concentrations than the NorESM integrated solver, while NorESM simulates higher column integrated concentrations close to the north-west coast of Australia and over Malaysia.

These comparisons show that the convection scheme can be fundamental in large-scale dispersion and may overshadow the influence of other possible sources of discrepancy between the off-line and on-line transport schemes including numerical
diffusivity.

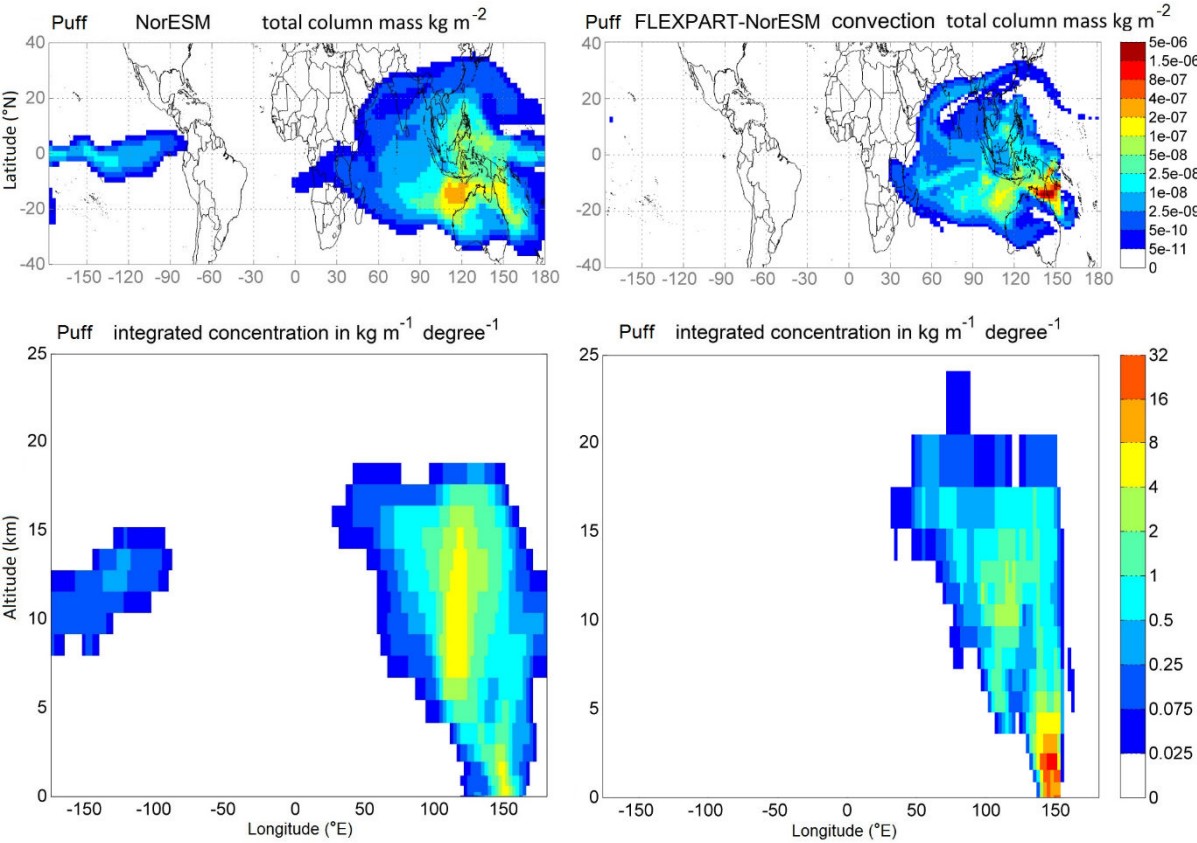

**Figure 5. Puff dispersion after 144 hours from the GUI source. NorESM integrated solver results (left panels) and FLEXPART-NorESM/CAM results (right panels) are shown. The upper panels show maps of vertically integrated tracer concentrations, the lower panels show meridionally integrated tracer concentrations as a function of longitude**
**and altitude.**





### 3.1.3    Insight in the sensitivity of the results to boundary layer height

FLEXPART and NorESM have a similar formulation for the definition of the boundary layer depth, both based on Vogelezang and Holtslag (1996), which uses the critical Richardson's number concept. In all the simulations, but one, of those listed in Table 1 the differences between the scalar transport in the two models could be explained in terms of the difference in numerical

5    diffusivity and/or deep convection scheme. In this section, we will investigate the exception where small differences in the height of the boundary layer top triggered visible differences in the scalar dispersion. FLEXPART has the option to activate a SGS parametrization of the topography effects that increases the boundary layer depth in presence of significant unresolved topography (see FLEXPART user guide). The role that the SGS parametrization of topography may have on the scalar dispersion in FLEXPART-NorESM/CAM is shown in Fig. 6 for a short duration (30 minutes) release from the ALE source.

10   The right and middle panels show the result from FLEXPART-NorESM/CAM without and with SGS topography respectively while the left panel shows the results for NorESM. Comparing the right and central panels, it is clear that in this case the SGS topography effect is substantial. It generates a higher boundary layer allowing the tracer to reach an altitude with different horizontal transport. The better consistency between NorESM and FLEXPART-NorESM/CAM with topographic effects is a bit surprising given that NorESM does not account explicitly for this effect.

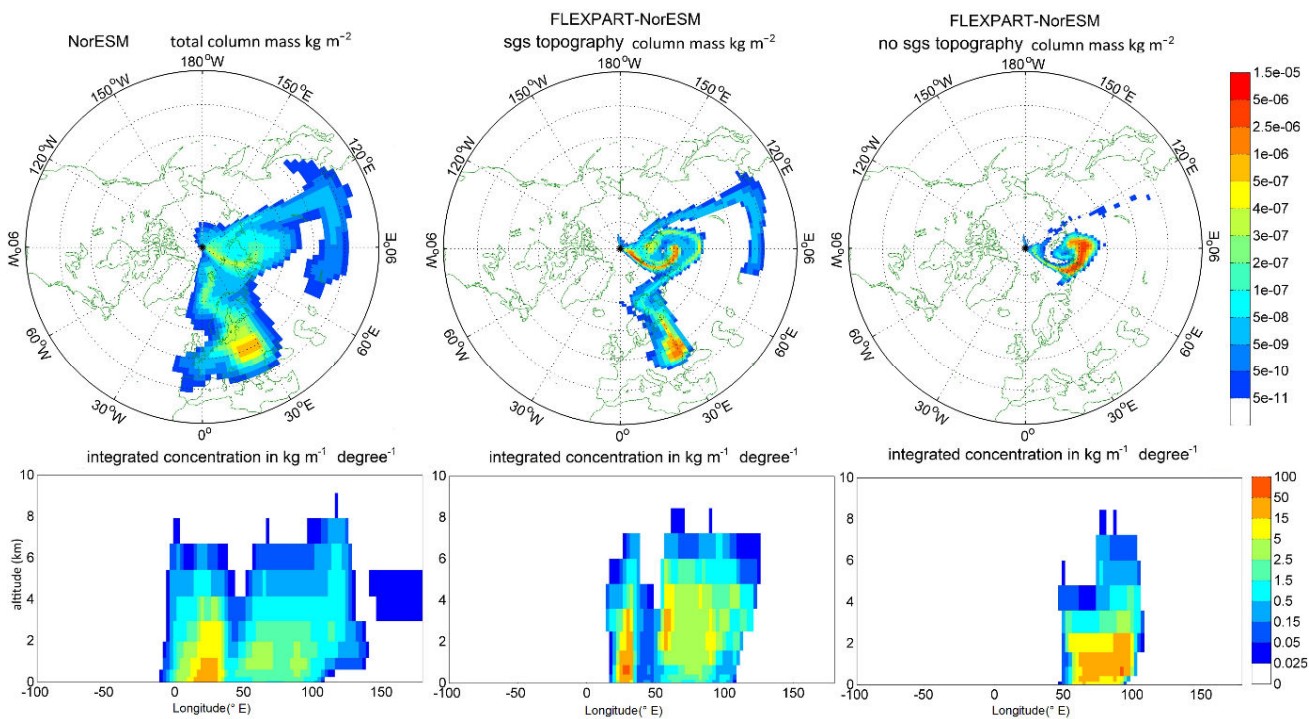

**Figure 6. Tracer dispersion for a 30 minutes release from the ALE point after 144 hours of simulated time, with the NorESM on-line tracer solver (left panels) versus Lagrangian tracking with FLEXPART-NorESM/CAM. FLEXPART results are shown with (middle) and without (right) the SGS topography PBL scheme; no SGS topography PBL scheme was used in NorESM (left).**



However, Fig. 7 shows the PBL depths in FLEXPART calculated when the dispersion of the puff, with and without the SGS parameterization of topography, initially start departing. The PBL depths seems in better agreement with those in NorESM if the SGS parametrization is used. This is visible along the coast of Greenland including the northeast of Greenland, where the puffs tracer distribution originally started deviating. This could be related to the coarser time resolution (3 h) of the

5   meteorological field used in FLEXPART-NorESM/CAM compared to the native resolution (30 minutes) of NorESM. The FLEXPART SGS parametrization of topography is indeed intended to also compensate the coarser time resolution. However, we note that in general the PBL depths calculated by FLEXPART with the SGS parametrization of topography activated are larger than those calculated in NorESM.

**Figure 7. Boundary-layer depths after 21 hours of simulation for FLEXPART-NorESM/CAM without parametrization of SGS topography (right), with parametrization of SGS topography (middle) and for NorESM (left). The red dot indicates the source location.**



### 3.1.4 Horizontal numerical diffusivity versus physical SGS diffusivity

In the sections above we have discussed the importance of the convection scheme and a case where the use of the SGS parametrization of topography played a role. However, in most cases the differences in the results between FLEXPART-NorESM/CAM and NorESM can be attributed to their different diffusivity. FLEXPART, as any Lagrangian particle model, is

only minimally affected by numerical diffusivity (see e.g. Reithmeir and Sausen, 2002). Therefore, in FLEXPART horizontal motions unresolved in the driving meteorological fields need to be parametrized as an additional horizontal diffusivity (see FLEXPART user guide). NorESM, on the other hand, does not have any parametrization of these effects and the horizontal diffusivity of the scalar is purely numerical. In the following test the puff release from the ALE point (see Fig. 6) and the plume from the SUM point (see Fig. 2, right panels) have been re-run with a tenfold increase in the horizontal diffusivity in

FLEXPART. The comparison shows that the results with the increased diffusivity in FLEXPART (Fig. 8) are considerably more similar to those of NorESM (Fig. 2 and 6, right panels). Albeit simple, this test shows that, for the resolution and numerical scheme used in NorESM, the horizontal diffusion in NorESM is about ten times stronger than the parametrized diffusion of unresolved scales used in FLEXPART. These results agree, at least qualitatively, with the finding of Reithmeir and Sausen (2002) where, for the ECHAM model, a better behaviour was found for their on-line Lagrangian transport scheme

compared to the semi-Lagrangian scheme (grid-based) due to an excessive numerical diffusion in the latter one. From the point of view of model inter-comparison (and FLEXPART-NorESM/CAM validation), it is reassuring that a large part of the difference between the two models can be explained in terms of just horizontal diffusivity. This comparison also shows the added value of the off-line Lagrangian model, as the numerical diffusion in NorESM is likely much too strong and leads to a too rapid destruction of filamentary tracer structures. By running FLEXPART-NorESM/CAM with NorESM output, such

structures can be recovered.





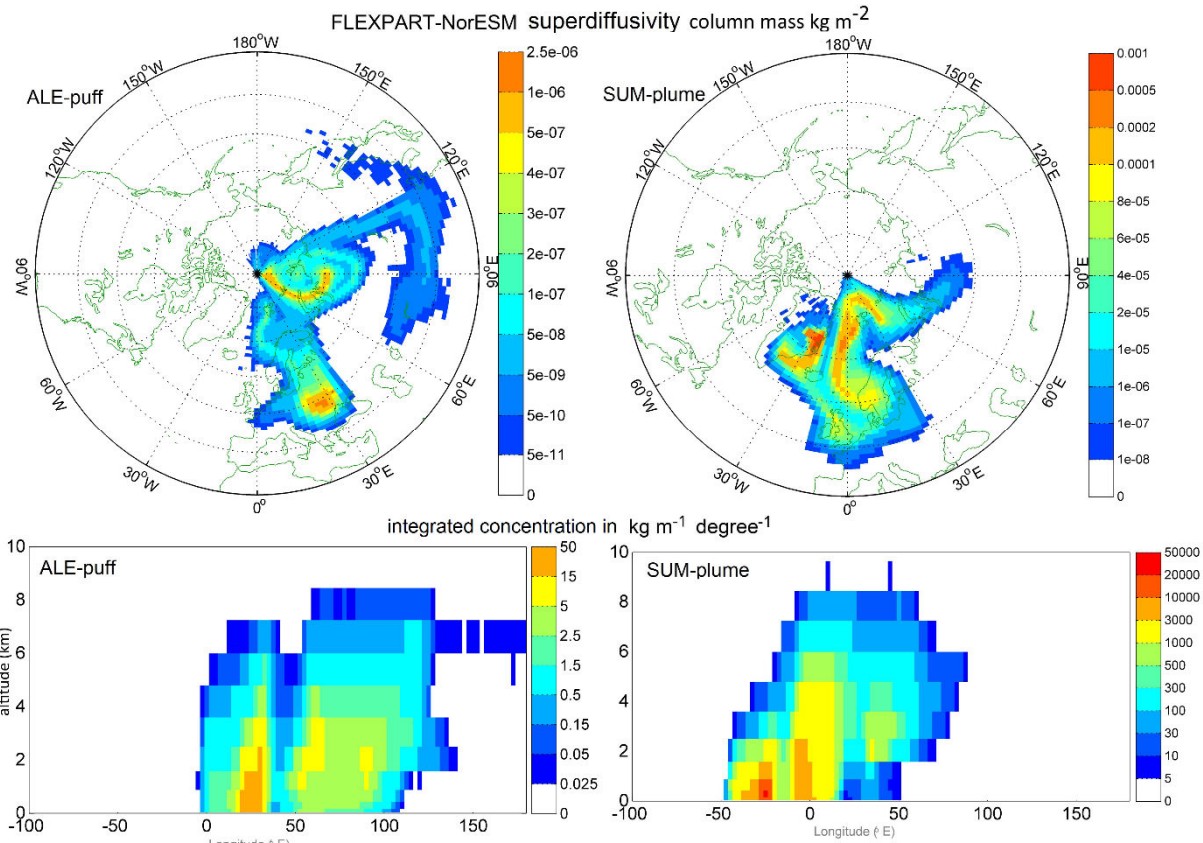

**Figure 8. Results for the 30 minutes release from ALE (puff, left panels) and the continuous release from SUM (plume, right panels) as simulated with FLEXPART-NorESM/CAM with a tenfold increase in the horizontal diffusivity coefficients. Compare with NorESM results in Fig. 6 and 2.**

## 3.2    FLEXPART-NorESM/CAM vs FLEXPART-ECMWF and the vertical well-mixedness of the tracer

In FLEXPART-NorESM the vertical velocity in the terrain following coordinates is obtained from the vertical velocity in pressure coordinates (omega) available in the NorESM output files. As shown previously, this does not introduce significant errors in short term simulations. However, in longer time simulations it may introduce errors in the vertical mass distribution that in Lagrangian stochastic modelling is often described as a perturbation of the vertical well-mixed state of the model (e.g. Thomson 1987). In order to test this, the well-mixed state in FLEXPART-NorESM/CAM and the standard FLEXPART-ECMWF model are compared. FLEXPART-ECMWF uses a different method to calculate the required vertical velocity in the terrain following coordinates which is based on the velocity in the original $\eta$ coordinate system rather than on $\varpi$. The standard FLEXPART model is here considered as a benchmark even though this model also does not perfectly conserve a vertically





well-mixed state over the whole atmosphere for long integration times. However, it is important to show that no significant additional deviations are introduced in FLEXPART-NorESM/CAM.

The well-mixedness of the tracer in physical space requires that the probability density function (pdf) of particle positions is proportional to the mean air density at any time if this was initially so (e.g. Thomson, 1987; Pope, 1987; Thomson, 1995; Stohl and Thomson, 1999; Cassiani et al., 2015). For a single atmospheric column with surface area $\Delta A (= \Delta x \Delta y)$ and focusing uniquely on the vertical direction this can be written in symbolic term as:

$$f_i(z_i, t) \propto \rho_i(z_i, t). \tag{9}$$

Here $\rho_i(z_i, t)$ is the local volume averaged air density as function of height and time and $f_i(z_i, t)$ is the pdf of the particle vertical position sampled over a discrete volume $\Delta V (= \Delta A \Delta z_i)$ around a specific vertical position $z_i$. Thus by initially spreading the particles according to (9) and assigning to the particle $n$ a fraction of the atmospheric mass, i.e. $m_n = M/N$, where $M$ is the total mass of the atmosphere and $N$ the total number of particle in the domain, it is possible at any time to calculate the mean atmospheric density inside a grid volume $V_i$ as

$$\rho_i(z_i, t) = \sum_{n=1}^{N_i} \frac{m_n}{V_i}, \tag{10}$$

where $N_i$ is the number of particles in the volume. This estimate converges to the true density while the number of particles increases to infinity and spatial discretization to infinitesimal, and it is affected by statistical noise and discretization. Thus, simple comparison of mean air density from the particles (from the Lagrangian model) and mean air density from the driving meteorological field (obtained by random sampling of the values inside a grid volume) allows testing of the well-mixed criterion and shows the level of consistency between the densities in the Eulerian and Lagrangian framework. If the model formulation was perfect, both densities should correlate perfectly, except for discretization errors and statistical noise. Four million particles were simulated for one year using the domain filling options of FLEPXART to model the evolution of the whole atmospheric mass in the domain. To reduce statistical noise we used a grid that changes longitudinal resolution from the equator towards the poles and has unevenly spaced vertical layers. More in detail, one grid cell extending for 360° is used near the poles while 36 grid cells are used at the equator and 11 unevenly spaced vertical layers, that are a subsample of the original FLEXPART terrain following layers, are used.



**Figure 9. Scatter plot of volume-averaged air density calculated from meteorological data of the driving model (NorESM and ECMWF), and using particle counting in FLEXPART according to Eq. (10). The linear regressions are shown as red lines.**

Figure 9 shows a scatter plot between the volume-averaged air density calculated as volume average of the driving model (using random samples) and using particle counts as reported in Eq. (10). The results are for vertical layers up to about 20 km. At the beginning of the simulation, the particles are initialized according to air density and any disagreement between air and particle density is due only to discretization and statistical noise. This is, fairly well, confirmed by the linear regression in Fig.



9, which gives $y = 0.97x + 0.01$  for the FLEXPART-ECMWF and $y = 0.97x + 0.01$ for FLEXPART-NorESM/CAM. Thus, slopes in both simulations are comparable and within 3% of unity. After one year of simulation the well mixedness can be altered by errors in the physical and numerical formulations (see e.g. Cassiani et al. 2015) both in FLEXPART-ECMWF and FLEXPART-NorESM/CAM.  The linear regressions are $y = 0.84x + 0.06$ for FLEXPART-ECMWF and  $y = 0.93x + 0.03$   for  FLEXPART-NorESM/CAM. In the test FLEXPART-NorESM/CAM performs somewhat better, which is satisfactory in terms of validation of FLEXPART-NorESM/CAM, but we do not have a good explanation for this better performance. However, we notice that the scatter of the values is larger in FLEXPART-NorESM/CAM. We speculate that this is due to the coarser resolution of the driving meteorological fields in NorESM introducing larger discretization errors. The relative behaviour between the two types of the FLEXPART model is the focus here. However, we note that a worsening of the initial well-mixed condition has to be expected since the FLEXPART runs used the fixed time step option, with a time step of 10-minutes and a refinement to 3 minutes for the vertical velocity component in the PBL. This option does not adapt the time step to the Lagrangian turbulent integral time scale when integrating the stochastic differential equations for the particle velocity in the PBL and, therefore, does not preserve perfectly the well-mixed state in the PBL. This option is typically used for global domain simulations (see also the FLEXPART user guide). An adaptive time-step option is also available that has better performance but is more computationally demanding. In addition, FLEXPART does not account for horizontal density gradients in its PBL formulation (see e.g. Cassiani et al 2015), and for any density gradient in the mesoscale fluctuations formulation. These approximations contribute to the deviation from the well-mixed state and therefore, to the inconsistency between the Eulerian and Lagrangian density fields.  This aspect may be important for passive tracers especially if there is a need to model continuously and for a long period the whole atmospheric mass in a domain. In general, consistency between the Eulerian and Lagrangian density fields is important if chemistry and mixing are included in a Lagrangian framework and it is fundamental if dynamically active scalars are involved in hybrid Eulerian-Lagrangian simulations (e.g. Muradoglu et al. 2001, Heinz 2003, Cassiani et al. 2007, Popov and Pope 2014, Grewe et al. 2014). We note that, for applications that do not require continuous particle trajectories, particles re-initialization procedures may be included to re-establish consistency between the Eulerian and Lagrangian density fields. However, these procedures should be avoided because they introduce the same errors that are inherent with semi-Lagrangian schemes (e.g. Reithmeir and Sausen, 2002).

## 4    Comparison of averaged transport patterns for six observatory locations in FLEXPART-NorESM/CAM, using FLEXPART-ECMWF driven by ERA-Interim fields

The previous comparisons have shown that FLEXPART-NorESM/CAM is technically working as expected. However, to test whether the combined modelling system FLEXPART with NorESM can also provide realistic transport climatologies, a further model inter-comparison between FLEXPART-NorESM/CAM and FLEXPART-ECMWF driven by ERA-Interim meteorology (Dee et al., 2011) was performed. This test also compares indirectly the climatologies of meteorological variables in NorESM and ERA-Interim that are important for driving FLEXPART (especially the wind data). For our comparison, we



chose one of the most common applications of global-scale Lagrangian particle models, the modelling of retroplumes (Stohl et al., 2003). Retroplumes from a measurement site, for instance, are often used to establish transport climatologies, to identify pollution sources, and to quantify source contributions to pollution at the site. If sources are at the surface, typically the FLEXPART output for the lowest model layer (so-called footprint) is of greatest interest. Therefore, the goal of this comparison

is to gain confidence in the performance of FLEXPART-NorESM/CAM  for generating footprint climatologies (see, e.g., Hirdman et al., 2010) by applying it to recent historical periods for which ERA-Interim data are also available.

In this section backward trajectories have been calculated with FLEXPART-NorESM/CAM and FLEXPART-ECMWF driven with ERA-Interim data for six observatories: ALE, BAR, SUM, TRO, ZEP (listed in Table 1) and Birkenes (BIR, 58º 23'N, 8º 15'E, 190 m a.s.l.). Each retroplume was simulated for 21 days backward in time. Retroplumes were calculated daily by

releasing 50000 particles at the observatory location for a period extending from beginning of December 1995 to end of December 2000. The output contains the calculated residence time (RT) for any day in seconds separately for 5 layers of the atmosphere (0-150 m, 150-1000 m, 1-3 km, 3-10 km, 10 km-top of domain). The gridded residence time (RT) output has a one-degree horizontal resolution.  Results are shown and discussed as five-year seasonal averages, for December, January and February (DJF), and June, July, August (JJA), for a layer of the atmosphere extending from 0-1 km, i.e. summing up the first

two layers.

FLEXPART has the possibility to treat physical loss processes of a tracer, such as wet and dry deposition. Therefore, we simulated both a passive tracer and a species resembling the behaviour of black carbon, which is subject to wet and dry deposition. Figures 10 and 11 show the results obtained for the conserved tracer and for four stations for JJA and DJF respectively. The residence time (in seconds) in the grid cell has been scaled by the grid surface area (in $km^2$) to avoid pattern

similarity trivially resulting from the reduction of the grid cell area with the latitude. The values were cut to a minimum averaged residence time of $10^{-4}$ second per $km^2$ and are expressed as the logarithm (base 10). It can be seen that the patterns for the averaged residence time of all the stations are very similar between FLEXPART-ECMWF and FLEXPART-NorESM/CAM, with extremely consistent differences between release locations and season. The results for the stations not shown here are comparable.




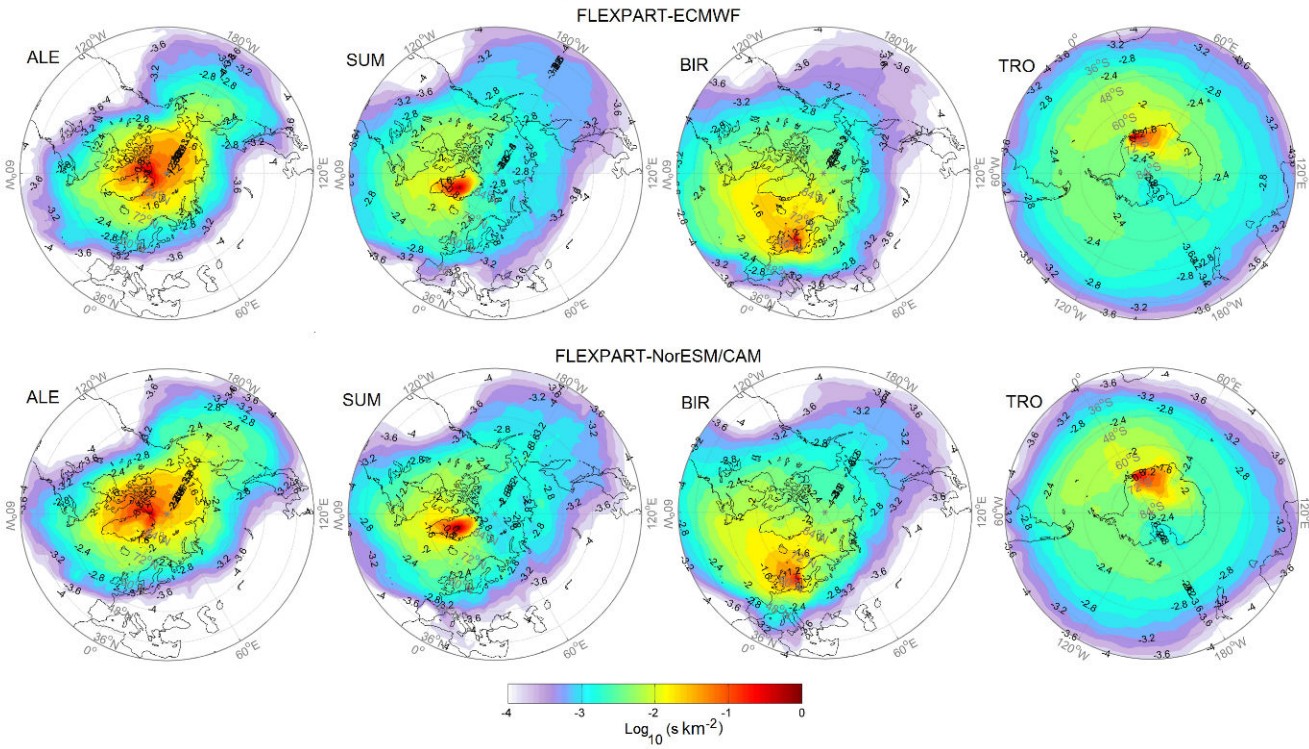

**Figure 10.** **Conserved tracer averaged residence time in the lowest 1 km of the atmosphere for 21-days retroplume, for four stations. Results are averaged over five years (1995-2000) for the months JJA. The values are expressed as the logarithm (base 10) of the residence time in seconds divided for the surface of the grid cell in km$^2$.**





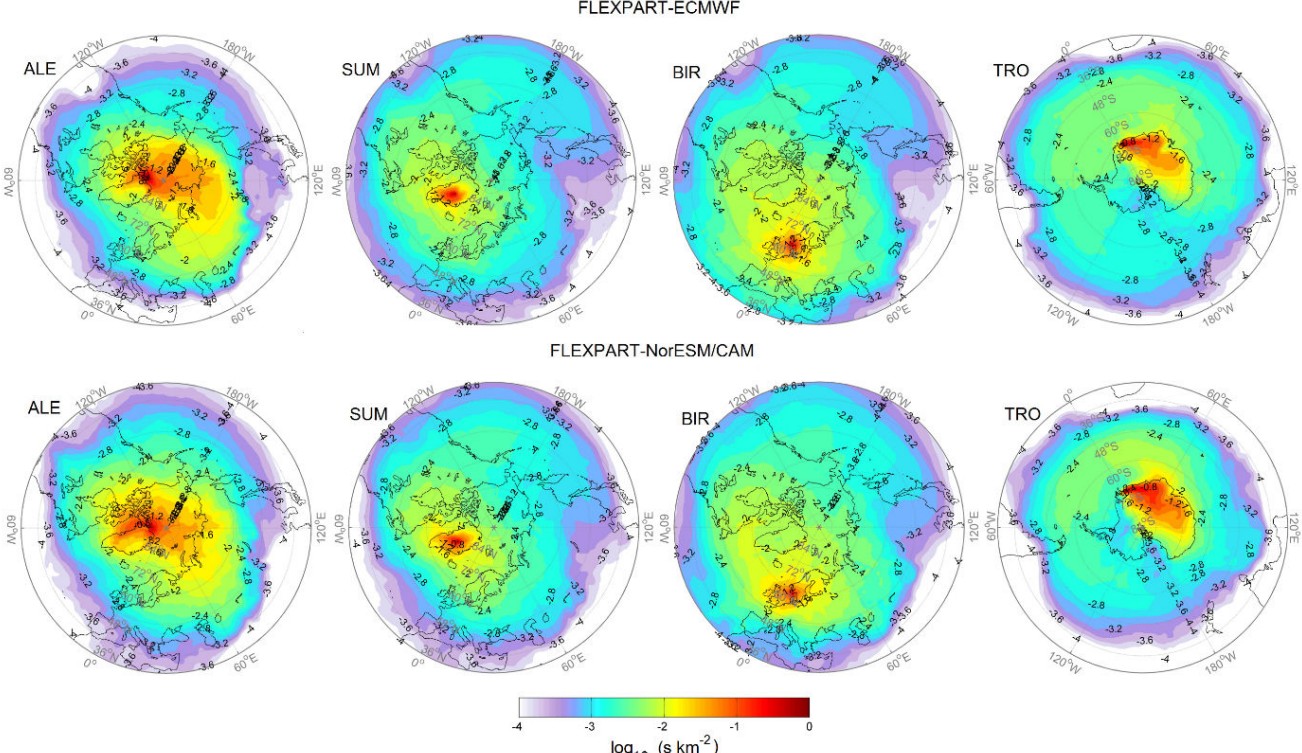

**Figure 11. As in Fig. 10 but for DJF.**

The residence times obtained for the black-carbon-like tracer subject to wet and dry deposition (not shown here) are marginally

less consistent between the two models. This is to be expected since differences in the precipitation patterns, especially, lead

to differences in wet deposition and thus tracer patterns. However, the overall picture is fundamentally unchanged and the

level of consistency between the results of the two models is still high. This can be seen in Table 2 which reports a statistical

comparison between the two models for both DJF and JJA and for both conserved and depositing tracer. A threshold value of

1 s for the averaged residence time in the grid cell is used. The Pearson correlation coefficient (Corr) and the fraction of data

within a factor of two (FA2) are shown. For the conserved tracer the correlation varies between 0.84 and 0.99 and FA2 values

are between 68% and 96%. For the depositing tracer, the correlation varies between 0.85 and 0.98 and FA2 values between

56% and 91%. The e-folding time scales resulting from the deposition processes are also shown in the table. Note that these

time scales are influenced by the local processes near the observatory as the retroplumes are calculated only from data for 21

days backward in time (see e.g. Cassiani et al. 2013). The time scales are in general shorter for the FLEXPART-NorESM/CAM

model in the northern hemisphere polar regions. In the Antarctic region (TRO) the time scales between the two models are

very similar.



| | Conserved tracer | | | | Depositing tracer | | | | e-folding time scale (days) | | | |
|---|---|---|---|---|---|---|---|---|---|---|---|---|
| | JJA | | DJF | | JJA | | DJF | | ECMWF | | NorESM | |
| | Corr | FA2 | Corr | FA2 | Corr | FA2 | Corr | FA2 | JJA | DJF | JJA | DJF |
| ALE | 0.92 | 82% | 0.84 | 85% | 0.95 | 77% | 0.85 | 77% | 11.7 | 21.8 | 7.6 | 17.3 |
| BAR | 0.89 | 86% | 0.88 | 68% | 0.87 | 56% | 0.9 | 62% | 10.9 | 20.0 | 8.4 | 13.2 |
| SUM | 0.99 | 92% | 0.96 | 91% | 0.98 | 79% | 0.93 | 85% | 14.0 | 11.1 | 10.3 | 10.2 |
| TRO | 0.9 | 86% | 0.88 | 77% | 0.9 | 71% | 0.86 | 84% | 14.1 | 21.2 | 15.2 | 22.0 |
| BIR | 0.93 | 89% | 0.94 | 92% | 0.93 | 90% | 0.93 | 90% | 13.4 | 9.8 | 12.4 | 8.3 |
| ZEP | 0.95 | 81% | 0.95 | 96% | 0.95 | 83% | 0.94 | 91% | 10.2 | 12.6 | 8.0 | 11.8 |

**Table 2. Comparison between the FLEXPART-NorESM/CAM and the FLEXPART-ECMWF (driven by ERA-Interim) results for the 5-year averaged residence time in the lowest 1000 m of the atmosphere. Correlation (Corr) and fraction of data within a factor of two (FA2) for JJA and DJF are reported for both the conserved and depositing (black carbon) tracer. The e-folding (deposition) time scales are also reported.**

## 5 Summary and future development

We have developed a version of FLEXPART, FLEXPART-NorESM/CAM, that can ingest input data from the NorESM1-M/CAM4-Oslo model. This provides a new advanced tool to directly analyse and diagnose atmospheric transport properties of the climate model NorESM in a reliable way and can be used for climatological studies. To validate this newly developed FLEXPART model version we performed multiple comparisons both with the on-line transport scheme embedded in NoreESM and both with the reference FLEXPART version. From these comparisons, we can draw the following conclusions:

- Comparison between on-line tracer calculations with the grid-based (vertically semi-Lagrangian) advection scheme built into NorESM1-M and the off-line Lagrangian particle tracer calculations in FLEXPART-NorESM/CAM showed very good agreement, even for releases over high terrain where errors introduced by the transformation of the vertical coordinate system are largest. In fact, in most cases the largest differences between the off-line Lagrangian and on-line grid-based transport schemes were attributed to the higher numerical diffusion in the on-line finite volume transport scheme. This was proven by artificially enhancing the diffusion in the Lagrangian model calculations by one order of magnitude, which resulted in much closer agreement between the two methods. This proves a possible added value of dispersion calculations for non-reactive tracers done with FLEXPART-NorESM/CAM compared to on-line tracer calculations.

- NorESM1-M/CAM4-Oslo and FLEXPART-NorESM/CAM use different convection schemes. Nevertheless, the tracer transport is similar both with the on-line and LS off-line transport schemes, even for release locations strongly



affected by deep convection. Conversely, switching off the convection scheme in FLEXPART-NorESM/CAM leads to very large differences from both on-line and LS transport using the convection scheme.

- Tests have also been performed on the vertical well-mixedness of particles in longer-term (1 year) simulations. While some deviations from well-mixedness occur, these are of a similar magnitude as in FLEXPART-ECMWF, confirming that the vertical coordinate transformation in FLEXPART-NorESM/CAM does not introduce additional errors. However, the deviations from well-mixed state may be relevant for some applications.

- A further model inter comparison was done for one of the most common applications of off-line Lagrangian models: backward in time calculations of the retroplume and the footprint emission sensitivity for specific measurement stations. The inter-comparison between FLEXPART-NorESM/CAM and FLEXPART-ECMWF driven by ERA-Interim meteorology showed that FLEXPART-NorESM/CAM provided realistic transport climatologies for the years 1995-2000. This lend confidence in using this combined tool for climatological analyses.

The current FLEXPART-NorESM/CAM model is based on the FLEXPART 9.1 model, which is a purely serial code. Lagrangian one-particle models as FLEXPART are well suited for trivial parallelization by running multiple instances of the same simulation with a different independent random number string. However, a parallel (MPI based) version of FLEXPART has been recently developed which can automatize parallelization. Therefore, we aim to include the parallelization in the FLEXPART-NorESM/CAM model branch in the near future.

## 6   Code and data availability

The FLEXPART-NorESM/CAM model branch can be downloaded from www.flexpart.eu. Any FLEPXART code is free software distributed under the GNU General Public License and it is maintained using the Git system. Detailed instructions for the downloading, installation and testing procedure are available at the link: www.flexpart.eu/wiki/FpClimateNorESM. To test the installation a subset of the NorESM meteorological fields, used here to drive the model, is provided at the link above. The complete set of meteorological fields is available upon request, including the 1990-2070 climate simulations. The model results discussed here are also available upon request.

## 7   Appendix A. Description of the NorESM and CAM4-Oslo models

This study uses the CMIP5-version of NorESM used for concentration-driven greenhouse gas experiments, NorESM1-M, which is thoroughly presented by Bentsen et al. (2013) and Iversen et al. (2013). It belongs to the family of state-of-science global climate and earth system models that contributed to the Coupled Model Intercomparison Project Phase 5 (CMIP5; Taylor et al., 2012). The basis for NorESM1-M is the Community Climate System Model (CCSM4; Gent et al., 2011). The ocean component in CCSM4 is replaced by a different ocean model (NorESM-O) which is an elaborated version of the Miami Isopycnic Coordinate Ocean Model (MICOM; Bleck et al., 1992) adapted for multi-century simulations in coupled mode by





Assmann et al. (2010) and Ottera et al. (2010). Further extensions are described by Bentsen et al. (2013). The atmospheric component is CAM4-Oslo and includes an advanced aerosol-cloud-chemistry representation (Seland et al., 2008; Kirkevåg et al., 2013) into the version of CAM4 included in CCSM4 (Neale et al., 2010). The land and sea-ice components – the Community Land Model (CLM4; Lawrence et al. 2011) and the Los Alamos Sea Ice Model (CICE4; Holland et al. 2012) –

are the same as in CCSM4, except that aerosol deposition on snow (Flanner and Zender, 2006) and sea ice are treated prognostically in NorESM1-M, and minor adjustments of parameters for the thermodynamic properties of snow on sea-ice are made.

With respect to physical processes in the atmosphere not directly associated with the aerosols, CAM4-Oslo applies the standard configuration of CAM4. This includes the Rasch and Kristjansson (1998) scheme for stratiform cloud processes and the CAM-

RT radiation scheme. Deep convective clouds are parameterized following Zhang and McFarlane (1995) extended with the plume dilution and convective momentum transport also used in CCSM4 (Richter and Rasch, 2008; Neale et al., 2008). Shallow convection follows a parameterization by Hack (1994). The turbulence parameterization includes computation of diffusivities for the free atmosphere, based on the gradient Richardson number, and an explicit, non-local atmospheric boundary layer parameterization (Holtslag and Boville, 1993). The finite volume (FV) dynamical core with a semi-Lagrangian

approach with re-interpolation to the grid for the vertical (Rasch et al., 2006; Lin, 2004) is used for transport calculations. The time integration within the FV dynamics is fully explicit, with sub-cycling within the 2D Lagrangian dynamics. However, the transport for tracers can take a larger time step (e.g., 30 minutes) as for the physics. The horizontal resolution is 1.89° in latitude by 2.5° in longitude, and there are 26 levels in the vertical with a hybrid sigma-pressure co-ordinate and model top at 2.917 hPa.

Two types of simulations have been used for adapting and testing FLEXPART with NorESM1-M and CAM4-Oslo. In a first experiment, the period 1990–2070 has been simulated (scenario RCP6.0 from year 2005 onward) using the fully-coupled NorESM1-M. The time-evolution of global mean atmospheric concentrations of the greenhouse gases $CO_2$, $CH_4$, $N_2O$, CFC-11 and CFC-12, the stratospheric concentrations of particulate $SO_4$ due to eruptive volcanoes, and the evolution of the total solar irradiance are all prescribed. Also the distribution of ozone is prescribed. Emissions of anthropogenic aerosols and their

precursors followed Lamarque et al. (2010) for the historical period, and ECLIPSE version 4a from 2005 onwards (Stohl et al., 2015). From this simulation, only results for the years 1999-2005 have been used in the present study, although the full period (1999-2070) has been actually used for FLEXPART simulations (results not shown here). A second experiment, designed for specific evaluation of the transport diagnostics by FLEXPART, followed the distribution of an inert tracer released from point sources during one month. This experiment used CAM4-Oslo as a stand-alone model instead of the fully coupled

NorESM1-M. The prescribed data ocean and sea-ice modules of CCSM4 were coupled to CAM4-Oslo, by which the sea surface temperature (SST) and sea-ice extent are prescribed as monthly fields derived from observations for year 2011 (Rayner et al., 2003). Present day values for the concentration of the greenhouse gases $CO_2$, $CH_4$, $N_2O$, CFC-11 and CFC-12 are prescribed.



We emphasize that FLEXPART-NorESM/CAM does not depend on any NorESM1-M specific features and can be driven with output generated by either NorESM or the Community Earth System Model (CESM; Lindsay et al. 2014), the successor of CCSM4. However, CCSM4 itself lacks the capability to output ten meter wind speed, and a code modification is therefore needed in order to produce output suitable for FLEXPART-NorESM/CAM. The following instantaneous 3-hourly fields need

to be specified in CAMs input namelist: PS:I, U:I, V:I, OMEGA:I, T:I, Q:I, CLDTOT:I, U10:I, TREFHT:I, PRECL:I, PRECC:I, SHFLX:I, TAUX:I, TAUY:I, QREFHT:I, SNOWHLND:I, FSDS:I. Here ":I" means instantaneous field and a full description of the field is give in Appendix B, Table 1B.

## 8   Appendix B

The FLEXPART-NorESM/CAM model runs off-line based on NorESM1-M and CAM4-Oslo output fields and uses the same

directory and input file structures as FLEXPART V9.1, which is described in detail in the FLEXPART user guide. We briefly remind that a so-called pathnames file is expected by the executable file in the same directory level where the executable file resides. This file points to directories where FLEXPART input data are provided. In FLEXPART-NorESM/CAM another path is added in the pathnames file pointing to the file "\grid_atm.nc" that defines the grid structure of the meteorological input fields. Contrary to the standard FLEXPART, in FLEXPART-NorESM/CAM, nesting of meteorological input data is not

possible and a no-leap calendar is used.

New Fortran routines reading the meteorological fields have been written since NorESM and CAM produce output in the Network Common Data Format (NetCDF) (e.g. Brown et al., 1993 and http://www.unidata.ucar.edu/software/netcdf/docs/index.html) instead of the Gridded Binary (GRIB) format (http://www.wmo.int/pages/prog/www/DPS/FM92-GRIB2-11-2003.pdf) format used by ECMWF/NCEP and the standard

FLEXPART model. In this, the present model version is similar to, and partially adapted from, the FLEXPART-WRF model (Brioude et al., 2013, Fast et al., 2006). Table 1B gives a complete list of the variables loaded from the NorESM/CAM output files. All the time dependent variable are instantaneous.





| NetCDF name | | Description |
|---|---|---|
| Lat | | Latitude, 96 values are expected |
| Lon | | Longitude, 144 values are expected |
| PHIS | $m^2\,s^{-2}$ | Surface geopotential |
| LANDFRAC | | Land-sea mask |
| SGH | m | Standard deviation of orography |
| P0 | Pa | Reference pressure |
| hyai | | coefficient $A_K/P_0$ of the eta dot levels in Eq. (1) (interface levels) |
| hybi | | coefficient $B_K$ of the eta dot levels in Eq. (1) (interface levels) |
| hyam | | coefficient $A_K/P_0$ of eta dot levels in Eq. (1) (U, V ,T and OMEGA levels) |
| hybm | | coefficient $B_k$, in Eq. (1) (U, V, T and OMEGA levels) |
| PS | Pa | Surface pressure |
| U | $m\,s^{-1}$ | Horizontal wind east-west component |
| V | $m\,s^{-1}$ | Horizontal wind north-south component |
| OMEGA | $Pa\,s^{-1}$ | Vertical wind component |
| T | K | Temperature |
| Q | $kg\,kg^{-1}$ | Specific humidity |
| CLDTOT | fraction 0-1 | Total cloud cover |
| U10 | $m\,s^{-1}$ | Ten meter wind velocity; components are obtained using TAUX and TAUY |
| TREFHT | K | Two meter temperature |
| PRECL | $m\,s^{-1}$ | Large scale stable precipitation rate (liquid + ice) transformed in ($mm\,h^{-1}$) |
| PRECC | $m\,s^{-1}$ | Convective precipitation (liquid + ice) transformed in ($mm\,h^{-1}$) |
| SHFLX | $W\,m^{-2}$ | Sensible heat fluxes |
| TAUX | $N\,m^{-2}$ | Surface stress east-west |
| TAUY | $N\,m^{-2}$ | Surface stress north-south |
| QREFHT | $kg\,kg^{-1}$ | Specific humidity reference height used to obtain dew point |
| SNOWHLND | m | Water equivalent snow depth |
| FSDS | $W\,m^{-2}$ | Down welling solar flux at surface used for stomata opening calculation |

**Table 1B. List of fields loaded from the NorESM/CAM output.**

**Acknowledgements**

This work has been supported by the Research Council of Norway through the projects EarthClim and EVA (grant no. 229771),

5 by Nordforsk through the Nordic Centers of Excellence ESTICC (grant no. 57001) and the project CRAICC, and through the European Commission FP7 projects PEGASOS (FP7-ENV-2010-265148) and ACCESS (FP7-ENV-2010-265863). Computational and storage resources for NorESM simulations have been provided by NOTUR (nn2345k) and NorStore (ns2345k).

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
