# Peer review of "The off-line Lagrangian particle model FLEXPART-NorESM/CAM (V1): model description and comparisons with the on-line NorESM transport scheme and with the reference FLEXPART model."

_Geoscientific Model Development, 2016_

## Referee Comment (RC1) · Anonymous Referee #1 · 30 Jun 2016

The paper describes a modified version of the FLEXPART model, which is able to use input data from the NorESM/CAM model. For this purpose, the vertical velocity, dew point and 10m wind speed have to be determined in NorESM/CAM and the input routines from FLEXPART have been modified to be capable of reading the NETCDF output of NorESM/CAM. The latter modifications however, are not discussed in detail, but the corresponding code is freely available for download from the FLEXPART repository.

The modification of NorESM are minor, delivering the required additional diagnostic

output; however, it is not obvious why the NorESM/CAM NETCDF data cannot be easily converted to GRIB format using classical data conversion tools (e.g. CDO). Nevertheless, the capabilities of this tool might prove useful, even though in my opinion most often FLEXPART will be driven by re-analysis data instead of using a chemistry-climate modelling system. Potential benefits which could be mentioned in the manuscript might be transport analysis for periods where no re-analysis data might be available, e.g. paleoclimate analysis and future climate projections.

Besides the technical development, the manuscript describes an evaluation of the new scheme, especially in comparison with the on-line transport of NorESM/CAM. The authors show that the re-calculation of the vertical velocity as needed by FLEXPART is realistic and that the two presented approaches yield similar results. On the considered scales, i.e. larger than 1° grid space, it is not surprising that the expression assuming hydrostatic balance is sufficient, as on these scales the hydrostatic approximation is highly realistic. A typical result from the comparison is that the lagrangian simulation results are far less dispersive compared to the Eulerian approach. However, this finding is well known and documented in the literature. In addition to the effect of the vertical velocity calculation the impact of the different convection schemes is analysed. This is in my opinion highly critical, even though not restricted to FLEXPART-NorESM, but to FLEXPART itself. The results show that for convectively dominated regions the impact of the selected convection parameterisation (based on different formulations of mass flux approaches) is substantial, as e.g. already discussed by Tost et al. (ACP, 2010). What is highly critical in my opinion is the application of a convection parameterisation on the output of a model which already includes a convection scheme. As the internal convection scheme of the driving model stabilises the atmosphere, the effect of the convection scheme is already included in the large-scale atmospheric state. The convective transport of tracers should therefore be determined from the convective massfluxes of the driving model, and not be re-calculated. Due to the application of the convection scheme in the driving model, CAPE, which is often used in the closure assumptions of the convection parameterisations (as also for the

Emanuel scheme), is substantially depleted. Furthermore, large-scale clouds which are calculated after convection furhter stabilise the atmosphere. On the other hand radiative and advective for moisture and temperature processes tend to de-stabilise the atmosphere which can trigger new convection. As FLEXPART does not calculate moisture and temperature advection or radiation the convective massfluxes from FLEXPART will be mostly underestimate the actual convection. The application of the Emanuel scheme (which tends to produce stronger convection (personal experience), compared to other schemes) might nevertheless lead to stronger convection than the Zhang-McFarlance-Hack scheme combination. The convective outflow height is highly critical for the vertical transport of tracers, as the large-scale vertical velocities in the upper troposphere are very low compared to the convective velocities. Even more difficult might be an overshooting into the lowermost stratosphere, where otherwise transport times might be on the order of weeks to months compared to individual almost instantaneous convective events. Therefore, in my opinion the application of a traditional convection scheme under these conditions should be discussed in detail, as well as further emphasize should be placed on the comparison of tracer transport into the upper troposphere by convection in the lagrangian approach.

Overall, I think that the manuscript is solid, but does not provide interesting new findings or developments, but could work as a documentation for the new model combination. How often this configuration will be applied and therefore the overall value of this manuscript, is still to be shown.

Minor remarks: To what kind of grid are the tracers mapped from the FLEXPART/NorESM model? Is this the same as the on-line NorESM grid? In Fig.1 the smallest grid size appears to be smaller, which could also lead to the impression that the eulerian approach is more dispersive (which is nevertheless true!).

Are you really sure, that NorESM does not include a subgrid-scale orography parameterisation? I am not an expert with this model, but as far as I know several GCMs, e.g. ECHAM, use such a scheme to account for unresolved topographical effects for both

boundary layer as well as gravity wave parameterisations.

In Fig.7 the continents are not (well) visible. Furthermore, the longitude, latitude coordinates mix with the values in the graph. A better resolved color scale could solve the issue, since then the values in the plot could be omitted.

Do you have any idea, what causes the "clustering" in the density distribution (Fig.9), which appears to be stronger in FLEXPART/NorESM than in FLEXPART/ECMWF, and appears ot be already present in the initial distribution?

Is NorESM nudged to the observed/(re-)analysed meteorology for the footprint simulations? Or is the climatology so independent on the individual weather state, that no significant deviations from ERA-INTERIM and the climatology can be detected?

Page24, line 10: NoreESM → NorESM
* * *

---

## Referee Comment (RC2) · Anonymous Referee #2 · 18 Jul 2016

This paper documented the details of on-line NorESM to FLEXPART. A comprehensive comparison with an off-line model is presented. Overall it is a very nice paper covering several different aspects of the Lagrangian stochastic dispersion model. One weakness is that it lacks the comparison with measurements or real data. Some actual measurements could shed more light on the model's performance.

For the on-line model, new routines were added for FLEXPART to read in and/or modify the NorESM outputs. It is a bit questionable that such modifications have enough

originality. However, the model inter-comparison provides some insights into the model and will be helpful for future users. The title and abstract could emphasize more on this part rather than the FLEXPART-NorESM/CAM(V1) itself.

The comparison between Lagrangian and Eulerian models in the first paragraph of the Introduction is somewhat biased. It is better to state that both models have their advantages and disadvantages.

Inline coupling of WRF and HYSPLIT, published in 2015 (shown below), is a very relevant work and could be mentioned here.

Inline Coupling of WRF-HYSPLIT: Model Development and Evaluation Using Tracer Experiments.this link opens in a new window Fong Ngan, Ariel Stein, and Roland Draxler, 2015. J. Appl. Meteor. Climatol., 54, 1162-1176. doi:http://dx.doi.org/10.1175/JAMC-D-14-0247.1

Appendix A is probably unnecessary here.

Specifics:

Abstract, "However, for both model versions there was some degradation ...": What degradation means here need to be specified.

Page 8, line 10: Please describe what the emission rate is and how many particles were released here?

Page 16, line 11, (Fig. 2 and 6, right panels): They should be the left panels.

Page 24, line 11: Remove "both" from "and both with the ..."

Page 28, table 1B: Add "Unit" at the top of column 2.

---

## Author Comment (AC1) · 10 Sep 2016

**Answer to reviewer n. 1**

We thank the referee for his review, his comments are reported below in Italic before our answers.

- *"... the input routines from FLEXPART have been modified to be capable of reading the NETCDF output of NorESM/CAM. The latter modifications however, are not discussed in detail, but the corresponding code is freely available for download from the FLEXPART repository. The modification of NorESM are minor, delivering the required additional diagnostic output; however, it is not obvious why the NorESM/CAM NETCDF data cannot be easily converted to GRIB format using classical data conversion tools (e.g. CDO)."*

Data conversion would mean that the data have to be stored twice in different formats. Given the large amounts of data produced by a NorESM run, this is a disadvantageous solution. We therefore prefer a model version that works directly with the native NorESM output format. This makes the use straightforward for any NorESM user and without double storing of data. Note that this is the same approach followed in the FLEXPART-WRF model version.

- *"Nevertheless, the capabilities of this tool might prove useful, even though in my opinion most often FLEXPART will be driven by re-analysis data instead of using a chemistry-climate modelling system. Potential benefits which could be mentioned in the manuscript might be transport analysis for periods where no re-analysis data might be available, e.g. paleoclimate analysis and future climate projections."*

Indeed, analyses of paleoclimate simulations and future climate projections are important potential applications. Perhaps we have not made this clear enough in the paper. Following the reviewer's suggestion, we made this clearer by addling a specific comment in the introduction (page 3, lines 9-11), reading: "The combination of the NorESM1-M climate model and FLEXPART allows Lagrangian transport analysis for periods where no re-analysis data might be available, e.g. paleoclimate analysis and future climate projections."

- *"Besides the technical development, the manuscript describes an evaluation of the new scheme, especially in comparison with the on-line transport of NorESM/CAM. The authors show that the re-calculation of the vertical velocity as needed by FLEXPART is realistic and that the two presented approaches yield similar results. On the considered scales, i.e. larger than 1_ grid space, it is not surprising that the express ion assuming hydrostatic balance is sufficient, as on these scales the hydrostatic approximation is highly realistic."*

We agree with the reviewer and we included a specific comment (page 6, line 19), reading: "We note that, at the considered scale, it is expected that the hydrostatic approximation is adequate"

- *"A typical result from the comparison is that the Lagrangian simulation results are far less dispersive compared to the Eulerian approach. However, this finding is well known and documented in the literature."*

Indeed, a typical result is that the off-line Lagrangian model compares similarly to the on-line Eulerian transport scheme but with the advantage of being far less dispersive, which we think is satisfactory. As the reviewer mention, the fact that Lagrangian numerical scheme are less diffusive is well known and some relevant references were already included in the manuscript in both the introduction and section 3.1.4. We now added two more relevant references (Sofiev et al. (2015) and Alam and Lin (2008)).

- *"In addition to the effect of the vertical velocity calculation the impact of the different convection schemes is analysed. This is in my opinion highly critical, even though not restricted to FLEXPARTNorESM, but to FLEXPART itself. The results show that for convectively dominated regions the impact of the selected convection parameterisation (based on different formulations of mass flux approaches) is substantial, as e.g. already discussed by Tost et al. (ACP, 2010). What is highly critical in my opinion is the application of a convection parameterisation on the output of a model which already includes a convection scheme. As the*

*internal convection scheme of the driving model stabilises the atmosphere, the effect of the convection scheme is already included in the large-scale atmospheric state. The convective transport of tracers should therefore be determined from the convective massfluxes of the driving model, and not be re-calculated. Due to the application of the convection scheme in the driving model, CAPE, which is often used in the closure assumptions of the convection parameterisations (as also for the Emanuel scheme), is substantially depleted. Furthermore, large-scale clouds which are calculated after convection furhter stabilise the atmosphere. On the other hand radiative and advective for moisture and temperature processes tend to de-stabilise the atmosphere which can trigger new convection. As FLEXPART does not calculate moisture and temperature advection or radiation the convective massfluxes from FLEXPART will be mostly underestimate the actual convection. The application of the Emanuel scheme (which tends to produce stronger convection (personal experience), compared to other schemes) might nevertheless lead to stronger convection than the Zhang-McFarlance-Hack scheme combination. The convective outflow height is highly critical for the vertical transport of tracers, as the large-scale vertical velocities in the upper troposphere are very low compared to the convective velocities. Even more difficult might be an overshooting into the lowermost stratosphere, where otherwise transport times might be on the order of weeks to months compared to individual almost instantaneous convective events. Therefore, in my opinion the application of a traditional convection scheme under these conditions should be discussed in detail, as well as further emphasize should be placed on the comparison of tracer transport into the upper troposphere by convection in the Lagrangian approach."*

We thank the reviewer for pointing out the interesting and comprehensive analysis by Tost et al. (2012). Interestingly, in their analysis Tost et al. (2012) found that both the Zhang and McFarlane (1995) and the Emanuel and Živković-Rothman (1999) convection schemes hardly calculate any convection in Polar Regions. This increases the confidence in our assumption (see section 3.1.1) that the two schemes behave similarly, and do not compute any convection, for the SUM source. Regarding this, an explicit reference to the work of Tost et al. (2012) is now included (page 10 lines 15-17), reading: "…In support of this assumption we note that Tost et al. (2012) found that both the Zhang and McFarlane (1995) and the Emanuel and Živković-Rothman (1999) convection schemes hardly calculate any convection in Polar Regions."
We agree with the reviewer that it would be better to obtain the convective mass fluxes directly from the driving model. However, these are not available from NorESM and it would be anyhow very demanding to store all these fluxes for later off-line calculation. To that extent, the convection parameterization in FLEXPART is a compromise to re-diagnose what has already been calculated in the dynamical model. This is still much better than ignoring these fluxes totally. However, this criticism about the convection scheme is general to FLEXPART and not specific to the present model version (as also stated above by the reviewer). For FLEXPART, Forster et al. (2007) studied this in detail, including re-diagnosis of convective precipitation and comparison of off-line and on-line calculated mass fluxes. They have shown that the differences are relatively small both for mass fluxes and precipitation, and most of the differences were attributed to the fact that different convection schemes were used in the on-line and off-line calculation, rather than to the on-line vs. off-line calculation per se. We do not think it is necessary to repeat this whole study for the present paper.

- *"Overall, I think that the manuscript is solid, but does not provide interesting new findings or developments, but could work as a documentation for the new model combination. How often this configuration will be applied and therefore the overall value of this manuscript, is still to be shown."*

We provide a new tool that allows Lagrangian analysis of the NorESM output. As also remarked above by the reviewer, this means that Lagrangian analysis are possible for periods where no re-analysis data might be available, like paleoclimate analysis and future climate projections. We performed a model validation, including models comparison, and we discussed in detail some advantages and limitations of global scale Lagrangian particles models. Concluding, we hope that this will be a useful tool for the scientific community and we agree with the reviewer that only time will show how much use this model version will see. However, we have early indications that some groups have started using the model version, and the current paper will be a valuable reference for all future uses of the model.

**Answer to Minor remarks:**

- *"To what kind of grid are the tracers mapped from the FLEXPART/NorESM model? Is this the same as the on-line NorESM grid? In Fig.1 the smallest grid size appears to be smaller, which could also lead to the impression that the eulerian approach is more dispersive (which is nevertheless true!)."*

Particles were sampled in $1° \times 2.5°$ (latitude $\times$ longitude) cells. This is now specified (page 8 line 25). We remind here that the grid is used here only to extract information from the particles and must be a compromise between resolution and number of particles sampled. We do not think that this influenced the comparison.

- *"Are you really sure, that NorESM does not include a subgrid-scale orography parameterisation? I am not an expert with this model, but as far as I know several GCMs, e.g. ECHAM, use such a scheme to account for unresolved topographical effects for both boundary layer as well as gravity wave parameterisations."*

NorESM has a turbulent mountain stress parametrization that includes standard deviation of orography, but it is turned off for the present simulations. Sub-grid scale standard deviation of surface orography is used in CAM's gravity wave drag parametrization. Therefore, sub-grid scale standard deviation of surface orography may influence PBL height.
However, what was written in the manuscript was probably not clear enough. What we meant is that in NorESM, the PBL height is diagnosed (on-line) without explicit additional accounting of sub-grid orography effects in the scheme that calculate the PBL height (this direct additional accounting is done in FLEXPART). We modified the text (page 15 lines 2-3, 6-7) to avoid misunderstanding.

- *"In Fig.7 the continents are not (well) visible. Furthermore, the longitude, latitude coordinates mix with the values in the graph. A better resolved color scale could solve the issue, since then the values in the plot could be omitted."*

The figure has been improved (see below). The continents are now clearly visible and the longitude and latitude coordinates do not mix anymore with the labels.

- *"Do you have any idea, what causes the "clustering" in the density distribution (Fig.9), which appears to be stronger in FLEXPART/NorESM than in FLEXPART/ECMWF, and appears to be already present in the initial distribution?"*

The clustering is due to the discretized nature of the volume average. Eleven vertical layers were used to discretize the atmospheric column. For a specific vertical level, the value may vary (depending on the longitude and latitude) but only within a limited range, this generates the clustering. Since NorESM has less vertical layers compared to ECMWF, it is likely that the volume average show a bit less variability. Therefore, data may appear more clustered in FLEXPART/NorESM compared to FLEXPART/ECMWF.

- *"Is NorESM nudged to the observed/(re-)analysed meteorology for the footprint simulations? Or is the climatology so independent on the individual weather state, that no significant deviations from ERA-INTERIM and the climatology can be detected?"*

NorESM simulations are not nudged.

- *"Page24, line 10: NoreESM ! NorESM"*

Fixed.

---

## Author Comment (AC2) · 10 Sep 2016

**Answer to reviewer n. 2**

We thank the referee for his review, his comments are reported in Italic before our answers.

- *"...Overall it is a very nice paper covering several different aspects of the Lagrangian stochastic dispersion model. One weakness is that it lacks the comparison with measurements or real data. Some actual measurements could shed more light on the model's performance."*

The present model has been developed for climatological simulations and we think that in this context the model comparison approach is a good way to validate it. Direct comparison with observations would be difficult, since the driving NorESM model is free-running and not made to reproduce a particular meteorological situation.

- *"...new routines were added for FLEXPART to read in and/or modify the NorESM outputs. It is a bit questionable that such modifications have enough originality. However, the model inter-comparison provides some insights into the model and will be helpful for future users. The title and abstract could emphasize more on this part rather than the FLEXPART-NorESM/CAM(V1) itself."*

We provide a Lagrangian model diagnostic tool for the NorESM climate model that can be used in e.g., future climate projections or paleoclimate studies. Moreover, it is our hope that the model tests and comparisons will provide a useful reference for future users of our model and more generally for users and developers of Lagrangian models. We think that the title is representative of the content of the paper.

- *"The comparison between Lagrangian and Eulerian models in the first paragraph of the Introduction is somewhat biased. It is better to state that both models have their advantages and disadvantages."*

We added, in line 1 of the introduction: "…and both modelling methods have their advantages and disadvantages".

- *"Inline coupling of WRF and HYSPLIT, published in 2015 (shown below), is a very relevant work and could be mentioned here. Inline Coupling of WRF-HYSPLIT: Model Development and Evaluation Using Tracer Experiments. Fong Ngan, Ariel Stein, and Roland Draxler, 2015. J. Appl. Meteor. Climatol., 54, 1162-1176. doi:http://dx.doi.org/10.1175/JAMC-D-14-0247.1."*

Thanks for pointing out this interesting paper. A reference has been included (page 8, line 5), reading: "An integrated solver is generally more consistent with the dynamics of the model than a post-processor (see e.g. Byun, 1999, Ngan et al., 2015) …". Notice, however, that FLEXPART-NorESM is not coupled on-line to NorESM.

- *"Appendix A is probably unnecessary here."*

We prefer to include Appendix A for completeness.

***Specifics:***
- *"Abstract, "However, for both model versions there was some degradation ...": What degradation means here need to be specified."*

We specified in the abstract: "…with the buildup of a bias and an increased scatter" (page 1, line 27).

- *"Page 8, line 10: Please describe what the emission rate is and how many particles were released here?"*

Results are shown for an emission rate of 1 kg s$^{-1}$. The number of particles used in the FLEXPART simulations was nine hundred thousand. The particle were emitted over a week for the continuous release (plume) and over 30 minutes for the almost instantaneous release (puff). Similar comments have been included in the paper (page 8, lines 25-27).

- *Page 16, line 11, (Fig. 2 and 6, right panels): They should be the left panels.*

Corrected (new page 17, line 1).

- *Page 24, line 11: Remove "both" from "and both with the ..."*

Corrected.

- *Page 28, table 1B: Add "Unit" at the top of column 2.*

Added.